# Synchronization of endothelial Dll4-Notch dynamics switch blood vessels from branching to expansion

Benedetta Ubezio[1,2†], Raquel Agudo Blanco[1,2†], Ilse Geudens[3,4], Fabio Stanchi[3,4], Thomas Mathivet[3,4], Martin L Jones[1,2], Anan Ragab[1,2], Katie Bentley[1,5*‡], Holger Gerhardt[1,2,3,4,6,7,8*]

[1]Vascular Biology Laboratory, London Research Institute, London, United Kingdom; [2]Lincoln's Inn Fields Laboratories, London, United Kingdom; [3]Vascular Patterning Laboratory, Vesalius Research Center, VIB, Leuven, Belgium; [4]Department of Oncology, Vascular Patterning Laboratory, Vesalius Research Center, Katholieke Universiteit Leuven, Leuven, Belgium; [5]Pathology, Beth Israel Deaconess Medical Center, Harvard Medical School, Boston, United States; [6]Max-Delbrück Center for Molecular Medicine in the Helmholtz Association, Berlin, Germany; [7]German Center for Cardiovascular Research, Berlin, Germany; [8]Berlin Institute of Health, Berlin, Germany

*For correspondence: kbentley@bidmc.harvard.edu (KB); holger.gerhardt@mdc-berlin.de (HG)

†These authors contributed equally to this work

Present address: ‡Computational Biology Laboratory, Center for Vascular Biology Research, Beth Israel Deaconess Medical Center, Harvard Medical School, Boston, United States

Competing interests: The authors declare that no competing interests exist.

**Abstract** Formation of a regularly branched blood vessel network is crucial in development and physiology. Here we show that the expression of the Notch ligand Dll4 fluctuates in individual endothelial cells within sprouting vessels in the mouse retina in vivo and in correlation with dynamic cell movement in mouse embryonic stem cell-derived sprouting assays. We also find that sprout elongation and branching associates with a highly differential phase pattern of Dll4 between endothelial cells. Stimulation with pathologically high levels of Vegf, or overexpression of Dll4, leads to Notch dependent synchronization of Dll4 fluctuations within clusters, both in vitro and in vivo. Our results demonstrate that the Vegf-Dll4/Notch feedback system normally operates to generate heterogeneity between endothelial cells driving branching, whilst synchronization drives vessel expansion. We propose that this sensitive phase transition in the behaviour of the Vegf-Dll4/Notch feedback loop underlies the morphogen function of Vegfa in vascular patterning.

## Introduction

The formation and maintenance of adequately branched and hierarchically organised networks of blood vessels is critical for all aspects of tissue growth and physiology (*Potente et al., 2011*). How the endothelial cells lining blood vessels collectively determine whether to form a new vessel branch or expand existing vessels is currently not understood. Similarly, the morphogenic and cellular principles underlying the chaotic vascular network formation in disease scenarios remain unclear. Recent work established that Dll4/Notch signalling between the activated endothelial cells functions to amplify stochastic differences in expression levels of the vascular endothelial growth factor receptors (Vegfr), ultimately establishing tip cells bearing high Vegfr expression, and inhibited stalk cells, bearing lower levels of signalling receptors (*Jakobsson et al., 2010*; *Hellström et al., 2007*; *Suchting et al., 2007*; *Phng and Gerhardt, 2009*; *Lobov et al., 2007*). The Dll4/Notch pathway thereby establishes a lateral-inhibition feedback loop with the Vegf-Vegfr pathway, controlling branching frequency by balancing the number of new tip cells with the number of stabilizing stalk

**eLife digest** Throughout life, blood vessels are constantly remodelled to ensure that oxygen and nutrients reach every part of the body where they are needed. If a tissue is not receiving an adequate blood flow, existing blood vessels may widen or new blood vessels may sprout from their walls. In certain diseases, such as cancer, blood vessels may grow excessively to form disorganized networks, and preventing this growth may help to treat these conditions. However, we do not fully understand how the body controls the size, shape and branching pattern of blood vessels.

For a new blood vessel to sprout out of an existing vessel, the tip of the new branch must first develop. The tip forms when the endothelial cells that line the blood vessel are activated by a protein called vascular endothelial growth factor A (Vegfa), which is produced by the surrounding tissue. The activated endothelial cells respond to Vegfa stimulation by producing the protein Dll4, which talks to neighboring endothelial cells to prevent them from also forming new tips. In a way, this process bears all the signs of a competition between cells, as they fight for which one is allowed to take the lead. The losers of this competition, when forced into subordination by the tips, also serve an important function, as they will help to form and elongate the base of the new sprout.

Although it is known that changes in the levels of Vegfa in tissues can cause blood vessel branching to alter dramatically, the mechanisms that enable this to occur are not well understood. Computer simulations of the process predicted that an unexpected synchronization of Dll4 dynamics would be triggered when Vegfa levels increased; however, this remained to be observed in real cells.

Ubezio, Blanco et al. have now used fluorescent markers to observe the Dll4 production in lab-grown mouse endothelial cells as they formed new vessel sprouts in response to Vegfa. This revealed that the levels of Dll4 fluctuate widely in individual cells. Time-lapse movies of the cells showed that as a new sprout forms, the levels of Dll4 in neighbouring cells fluctuate in an uncoordinated manner. However, increasing the amount of Vegfa in the cells indeed synchronizes these fluctuations. This causes the new sprout to retract and allows the original blood vessel to widen. Increasing the levels of Dll4 had the same effect.

Further experiments confirmed that increasing the amount of Vegfa also reduces blood vessel branching in tumours in mice by synchronizing the fluctuations in the levels of Dll4 in neighbouring endothelial cells. In the future, these results could help refine anti-cancer treatments that work by blocking the activity of Vegfa and Dll4.

cells (*Phng and Gerhardt, 2009*). Genetic mosaic experiments in 3D embryoid body sprouting assays and in vivo illustrated tip and stalk cells regularly change position as endothelial cells dynamically compete for the tip position (*Jakobsson et al., 2010*). Every 3–5 hr, the tip cell is exchanged, leading to continuously changing cellular neighbourhood relationships. Computational modelling of the feedback loop illustrated that it should suffice to pattern new sprouts, and comprise a robust mechanism for the establishment of a regular branching pattern (*Bentley et al., 2008*; *2009*). Recently integrated modelling and experiments further identified that this Notch/VEGF feedback drives differential VE-cadherin dynamics at individual endothelial junctions contributing to cell rearrangement behaviour (*Bentley et al., 2014a*). The meeting of new neighbours itself propagates iterative lateral-inhibition. Whether and how this activity might affect Notch signalling in individual cells over time remains unclear. In other cell systems, Notch signalling is highly dynamic, and Notch target genes oscillate both in single cells through autonomous feedback regulation, as well as in cell collectives (*Masamizu et al., 2006*). The latter is best studied during formation of somites from the presomitic mesoderm, in which Notch activity synchronizes intrinsic oscillatory gene regulation between neighbouring mesoderm cells (*Kageyama et al., 2007*; *Jiang, 2000*; *Hubaud and Pourquié, 2013*). In the absence of Notch signalling, cells eventually drift out of synchrony, finally leading to loss of somite patterning.

Dll4 over-expression in tumour cells in a glioblastoma model caused substantial vessel enlargement at the expense of branching (*Li et al., 2007*). Constitutive genetic endothelial activation of Notch4 leads to embryonic vessel diameter increase and arterio-venous shunt formation, further

suggesting that controlled Notch activity plays a role in vessel diameter regulation and adequate network remodelling (*Uyttendaele et al., 2001*). Collectively, existing studies show that reduced Notch signalling in endothelial cells promotes a highly branched network with small calibre vessels whereas increased Notch activity promotes a sparse network of large calibre vessels. Similarly, Vegfa shows strong dosage dependent effects on vascular patterning. Like *Dll4, Vegfa* is genetically haplo-insufficient, and overexpression causes embryonic lethality (*Miquerol et al., 2000*; *Carmeliet et al., 1996*). Surprisingly, despite the extensive body of work on Vegf and Dll4/Notch, our understanding of the principles and mechanisms that underlie these exquisitely dose sensitive effects on vascular patterning have hardly progressed beyond phenomenology. This may in part be because of the difficulties in analysing Vegf and Dll4/Notch signalling in a quantitative and dynamic manner, especially in vivo.

Here, we developed in vitro and in vivo analysis of Dll4 mRNA, protein and gene expression reporter dynamics under normal and pathological Vegfa stimulation, identifying a phase transition in the Dll4 dynamics that determines whether new vessels branch or expand. Computational modelling previously predicted that the Vegf-Dll4/Notch-Vegfr feedback loop normally establishes salt-and-pepper patterning between endothelial cells to regulate tip/stalk specification, but under elevated Vegfa levels, simulations predicted that this feedback loop would switch to drive the cells to collectively fluctuate their Dll4 levels in contiguous clusters, unable to stabilize into a heterogeneous pattern (*Bentley et al., 2009*). This highlights how the non-linear feedback involved in Vegf/Notch signalling can make it extremely hard to intuit how perturbation conditions, such as elevated Vegf, will impact on dynamics. Importantly, clear experimental evidence for the predicted dynamics and changing behaviours has been difficult to obtain. Further more, the computational models contain a limited parameter set, thus simplifying the complexity, potentially missing critical modifiers. Such modifiers may not only be molecular components, but also effects that originate from differences in cell shape and geometries, as these can trigger changes to signalling pathway dynamics (*Bentley et al., 2009*; *2014b*). In the present study, we therefore chose to combine and compare refined computational models that reflect the experimental assays and their endothelial geometries and integrate specific experimental assays and computational modelling throughout. Using high Vegfa levels in embryoid body assays, intraocular injection of Vegfa, the oxygen induced retinopathy model of ischemia driven ocular neovascularization, and finally syngenic mouse glioblastoma tumours, we present evidence for local Notch-dependent synchronization of Dll4 dynamics leading to vessel expansion whilst disrupting branching.

## Results

### *Dll4* levels fluctuate collectively rather than differentially under high Vegf in silico and in vitro

In order to gain first experimental insight into the dynamic behaviour of Dll4/Notch signalling under normal versus elevated Vegf conditions, we performed a time course experiment on endothelial monolayers. We collected mRNA from endothelial monolayers treated with either 50 ng/ml Vegfa 164 (normal) or 1 µg/ml Vegfa 164 (high) (*Figure 1e–i*). We monitored *dll4* mRNA levels by qPCR over a period of 9 and 24 hr post-stimulation. High Vegfa consistently induced fluctuations with high amplitude and several peaks (*Figure 1f,i*), which given the population based measurement indicates the cells are fluctuating in relative synchrony. Lomb-Scargle analysis (*Dequéant et al., 2006*) showed that the dominant periodicity in each dataset was 5–6 hr. The modest and varying degree of confidence in this analysis however suggests that these dynamic patterns in vitro are inherently noisy. Under normal Vegfa levels, *dll4* mRNA showed an unexpected low-amplitude rise and decline, but then remained relatively unchanged (*Figure 1e*). We had hypothesized these conditions should permit a stabilized salt and pepper pattern, manifested as a stable population level of *dll4*. To investigate whether this observation contests our current working model of Vegf/Notch feedback or if it can be explained simply by the different geometric nature of cells in a monolayer compared to a sprouting system impacting on the ensuing signalling dynamics, we simulated the Vegf/Notch feedback in a 2D monolayer geometry (see Materials and methods). This model revealed that the restricted ability of cells to change shape in a monolayer (without filopodia extensions and Vegf gradients) destabilizes the salt and pepper pattern; they instead fluctuate between salt-and-pepper

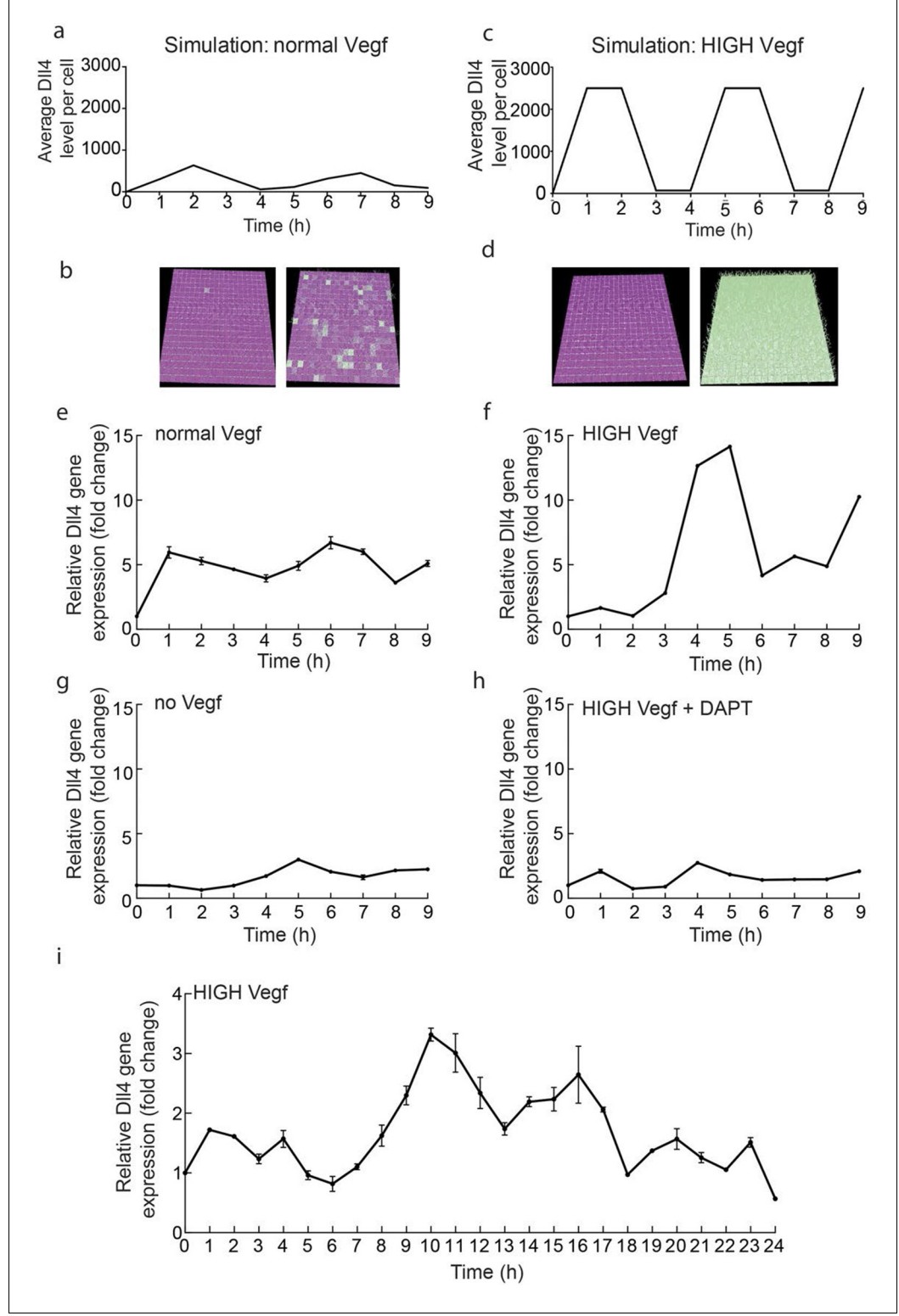

**Figure 1.** Endothelial Dll4 mRNA levels fluctuate dynamically in a Vegfa and Notch dependent manner. (**a–d**)
Simulation of Dll4 dynamics in a monolayer of 20×20 cells using the memAgent-Spring model under normal Vegfa
and high Vegfa (20x normal level). (**a**) The total Dll4 level across all cells fluctuated overtime with a regular
periodicity even though normal levels (**b**) generated a transient salt and pepper pattern with many non-adjacent
cells high in Dll4 at high points in the fluctuation and few in the low phase. (**c**) High Vegf caused the cells to
synchronise with the two phases of behavior showing either (**d**) most adjacent cells low, or most high in Dll4. Dll4

*Figure 1 continued on next page*

*Figure 1 continued*

level low to high are represented purple-green. (**e–i**) Quantitative real time PCR analysis of *dll4* mRNA levels in bEND5 cell monolayer. Cells were starved for four hours with serum-depleted medium and then stimulated with medium supplemented with either 50 ng/ml (**e**), 1 µg/ml (**f, i**), 0 Vegf (**g**), or 1 µg/ml Vegf and 50 µM DAPT (**h**). Cell lysates were collected every hour for the times indicated in the graphs. Values represent means ± S.D of technical replicates.

pattern-like states and then near-uniform Notch activation across the cells (*Figure 1a,b* and *Video 1*). Simulation of higher levels of Vegfa in this simple model led to high amplitude, synchronized oscillations of Dll4 (*Figure 1c,d* and *Video 2*).

To confirm that the fluctuations observed in vitro are indeed Notch regulated, we utilized the gamma-secretase inhibitor DAPT, a potent inhibitor of Notch signalling (*Hellström et al., 2007*), which completely abolished the fluctuations of *dll4* levels under high Vegfa (*Figure 1g,h*). Taken together these results suggest that high Vegfa levels synchronize contiguous endothelial cells in their fluctuating expression of *Dll4*, leading to dynamic and recurring fluctuations that require Notch activity.

## Development of a destabilised reporter to study individual cell Dll4 dynamics

To gain insight into single cell versus population dynamics of *Dll4* in vascular sprouting conditions, we generated a novel dynamic fluorescent reporter for *Dll4* expression (*Figure 2*). Given the lack of detailed knowledge on the transcriptional regulation of *Dll4*, we used a BAC clone containing the entire genomic locus of mouse *Dll4* (RP23_46P4, BACPAC CHORI) and replaced the stop codon in exon 11 with a viral self cleaving P2A peptide sequence (*Hsiao et al., 2008*), followed by a destabilized version of the yellow fluorescent protein Venus (*Li, 1998*; *Corish and Tyler-Smith, 1999*) and a selection cassette flanked by loxP sites by recombineering (*Yu et al., 2000*)(*Figure 2—figure supplement 1a,b*). We selected several ES cell lines carrying one, two or more copies of the transgene either replacing the endogenous *Dll4* allele (knock-in), or integrated in distant sites. Heterozygous knock-in ES cells and those carrying one distant copy of the *Dll4*-P2A-dVenus construct gave rise to high degree chimeras and germ line transmission, effectively establishing colonies of viable and fertile mice. *dll4* mRNA and protein levels appeared unaffected by the P2A-dVenus cassette (not shown). ES cells and mice carrying one additional copy showed expectedly 1.5 times the normal *dll4* mRNA levels (*Figure 2—figure supplement 1d*). Studying dVenus protein expression and Dll4 protein expression in ES cells and in mouse retinas from this strain showed perfect correlation between cells expressing dVenus and Dll4 protein (*Figure 2a–d, g, h*).

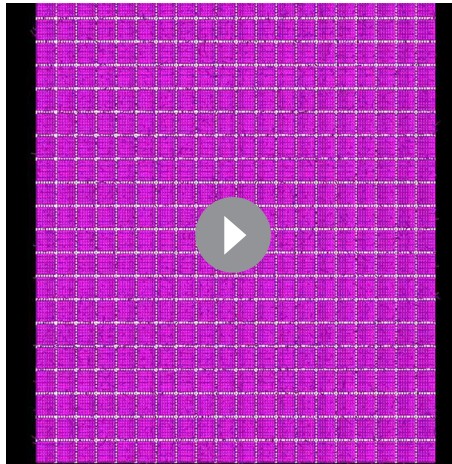

**Video 1.** Monolayer simulation normal Vegf.

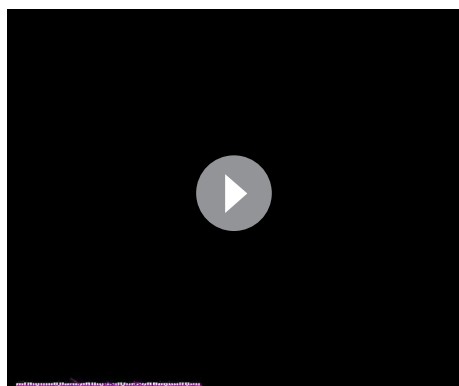

**Video 2.** Monolayer simulation high Vegf.

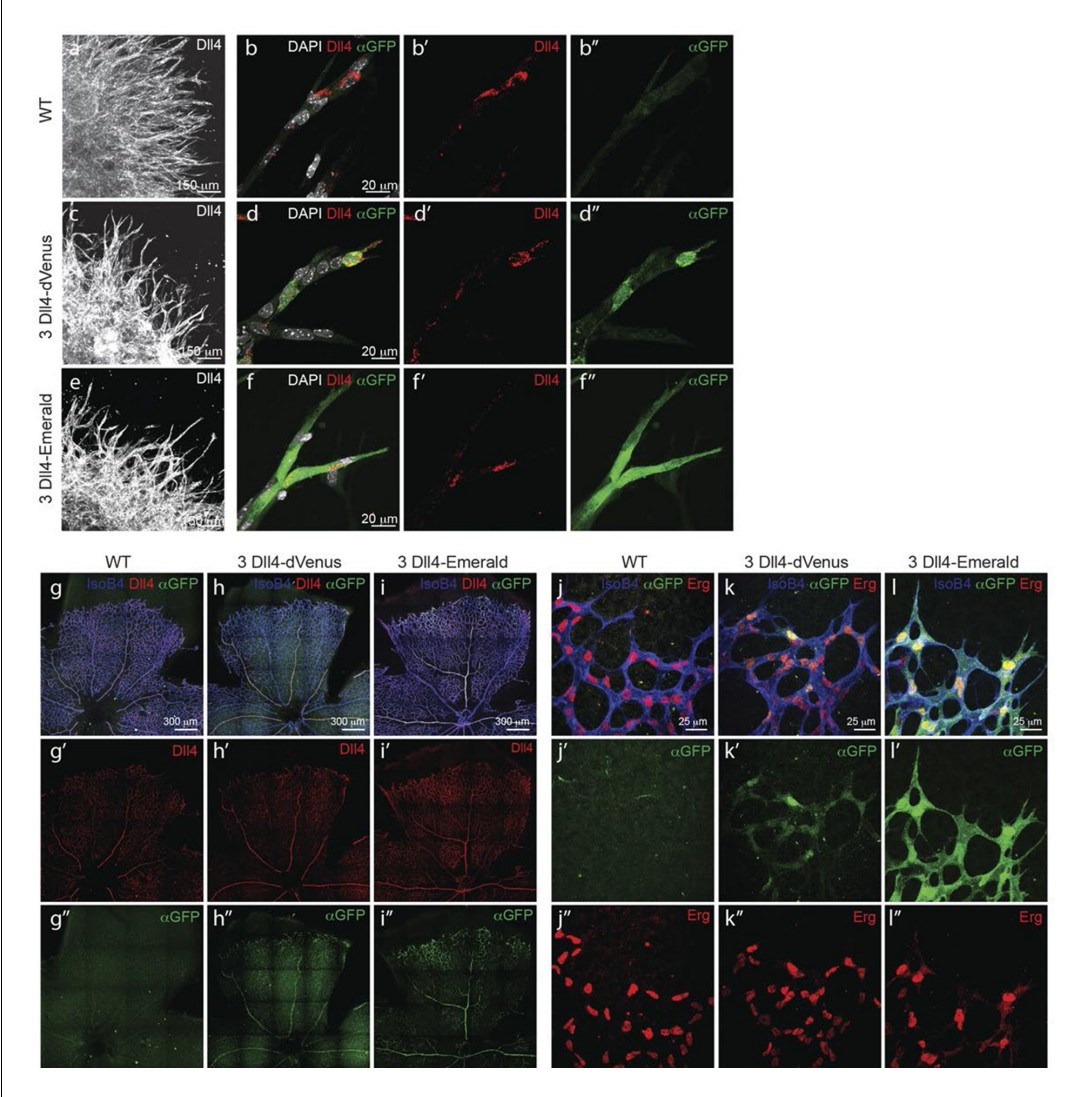

**Figure 2.** Dynamic and stable genomic reporters of Dll4 expression show differential distributions. (**a–f**) Representative confocal overview (**a, c, e**) and high magnification (**b, d, f**) images of vascular sprouts in WT (**a, b**), 3*Dll4*-dVenus (**c, d**), 3*Dll4*-Emerald (**e, f**) embryoid bodies immunolabelled with antibodies specific to Dll4 (white in **a, c, e** and red in **b, d, f**) and GFP (green). Cell nuclei labeled with DAPI (white; only shown in **b, d, f**). (**g–l**) Isolectin B4 (blue), Dll4 (red in **g–i**), anti-GFP (green) and endothelial nuclear marker ERG (red in **j–l**) protein staining in representative overview tile-scan (**g–i**) and high magnification (**j–l**) images of whole-mounted WT (**g, j**), 3*Dll4*-dVenus (**h, k**) and 3*Dll4*-Emerald (**i, l**) P5 retinas.

The following figure supplements are available for figure 2:

**Figure supplement 1.** Generation of dynamic dVenus and stable Emerald Dll4 reporters.

**Figure supplement 2.** (g–i) Quantification of sprout density (g), sprout diameter (h), and nuclei number (i) in WT, 3*Dll4*-dVenus and 3*Dll4*-Emerald embryoid bodies (See Materials and methods for details).

Using the same strategy, we also generated ES cells and mice carrying the more stable and thus less dynamic fluorescent reporter Emerald for *Dll4* (*Figure 2—figure supplement 1c*).

Both reporters have a fast maturation time (*Nagai et al., 2002*; *Day and Davidson, 2009*); however, dVenus has an extremely short half-life (15 min), while Emerald has a half-life of 24–26 hr (*Li, 1998*; *Corish and Tyler-Smith, 1999*). Like *Dll4*-P2A-dVenus, also *Dll4*-P2A-Emerald was expressed in sprouting endothelium, and generated ES cells and viable reporter mice with no

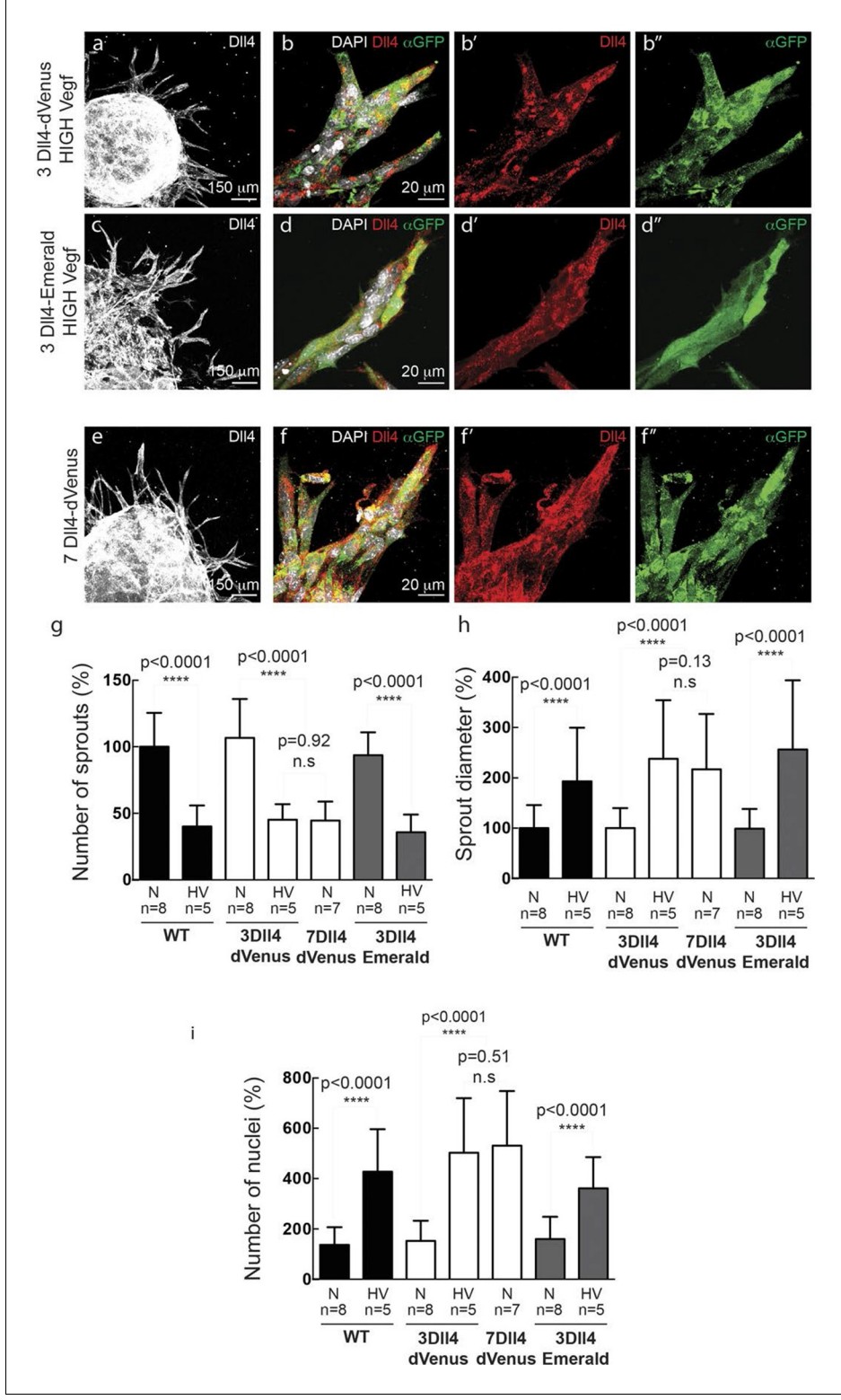

**Figure 3.** In vitro high Vegfa or Dll4 levels lead to a switch from branching and elongation to sprout expansion. (a-d) Dll4 (white in **a**, **c**; red in **b**, **d**) and reporter protein (anti-GFP, green) staining in representative confocal overview (**a**, **c**) and high magnification (**b**, **d**) images of vascular sprouts in 3*Dll4*-dVenus (**a**, **b**) and 3*Dll4*-Emerald (**c**, **d**) embryoid bodies cultured under high Vegf concentration. Cell nuclei are labeled with DAPI (white; only shown in **b**, **d**). (**e**, **f**) Representative confocal overview (**e**) and high magnification (**f**) images of 7*Dll4*-dVenus embryoid bodies immunolabelled with antibodies specific to Dll4 (white in **e**, red in **f**) and GFP (green). Cell nuclei labeled

*Figure 3 continued on next page*

*Figure 3 continued*

with DAPI (white; only shown in f). (g-i) Quantification of sprout density (g), sprout diameter (h), and nuclei number (i) in WT, 3*Dll4*-dVenus, 7*Dll4*-Venus and 3*Dll4*-Emerald embryoid bodies treated with either normal (N) or high Vegfa levels (HV). (See Materials and methods for details). Values represent means ± S.D. n= number of individual embryoid bodies analysed. P values calculated using two-tailed unpaired t-test.

The following figure supplements are available for figure 3:

**Figure supplement 1.** Simulated Dll4 dynamics with 2, 3 and 7 copies of Dll4.

**Figure supplement 2.** In vitro synchronization of cell signaling and behavior driven by *Dll4* overexpression is Notch dependent.

changes in sprouting, branching and retinal patterning (*Figure 2e–i*). However, unlike dVenus, the distribution of Emerald protein was more widespread than Dll4 protein or *dll4* mRNA. Both in the embryoid body ES cell sprouting assay (EB) (*Jakobsson et al., 2007*) and in the postnatal mouse retina, staining for dVenus showed the typical salt-and-pepper distribution pattern, with high levels in tip cells, and low levels in neighbouring stalk cells (*Hellström et al., 2007*). The long sprouts in EBs showed strong expression at the tip, followed by one or two cells with low expression, and then again single cells with high expression (*Figure 2d*). This pattern matches the principles of lateral inhibition thought to govern Dll4/Notch signalling in endothelial cells (*Hellström et al., 2007*; *Suchting et al., 2007*; *Phng and Gerhardt, 2009*; *Lobov et al., 2007*). Emerald however was present at high levels in both Dll4 protein positive and Dll4 protein negative cells (*Figure 2f*), suggesting that Dll4 protein half-life is considerably shorter than that of Emerald. Also in the retina, Emerald protein highlighted large stretches of the vascular front with little difference between neighbouring cells, illustrating that most cells at the front experienced a phase of *Dll4* expression, and that the reporter accumulates over time (*Figure 2i,l*). The salt-and-pepper distribution of dVenus and widespread distribution of Emerald provide the first evidence for dynamic phases of *Dll4* expression in sprouting angiogenesis in vivo. Remarkably, these results indicate that the salt-and-pepper distribution is not fixed, but dynamic, consistent with the idea of continued and iterative competition for the tip position via Dll4/Notch signalling (*Jakobsson et al., 2010*).

## Individual Dll4 dynamics synchronise under high Vegf and high Dll4 conditions

Given the known haplo-insufficiency in *Dll4* mutant mice (*Gale et al., 2004*), the homozygous viability and lack of vascular defects in mice and ES cells carrying the *Dll4*-P2A-dVenus or *Dll4*-P2A-Emerald alleles indicate that the targeted *Dll4* allele is fully functional. Furthermore, the distant integration leading to slightly higher Dll4 levels nevertheless fully recapitulates endogenous and functional *Dll4* expression patterns. Only the ES cell line, and mouse line, carrying an additional distant integration of *Dll4* (3*Dll4*) showed sufficient dVenus levels to allow detection without antibody staining. Therefore, we used this line to study the effects of normal and high Vegfa levels on endothelial cell behaviour and sprouting, and for dynamic studies of *Dll4* expression (*Figures 3*, *4*).

WT control EB and *Dll4*-dVenus EBs showed a highly branched network of long and slender sprouts under normal Vegfa conditions (*Figure 2a,c*). Dll4 protein and dVenus showed the typical salt-and-pepper distribution correlating with normal sprouting patterns (*Figure 2b,d*). However, high Vegfa conditions led to conspicuously short and stunted sprouts, with much wider diameter (*Figure 3a*). Interestingly, this morphological change from elongation to expansion invariably correlated with altered Dll4 and dVenus distribution. Although still most strongly expressed at the tip, Dll4 and dVenus were expressed at very high levels in clusters of neighbouring cells close to the tip (*Figure 3b*). Thus, as predicted by computational modelling (*Bentley et al., 2008*; *2009*) (*Figure 1c*), high Vegfa levels appear to disrupt the typical lateral-inhibition salt-and-pepper pattern and instead lead to more synchronized fluctuations of Dll4 levels in contiguous cells, thereby disrupting branching. Also the 3*Dll4*-Emerald ES cells responded to high Vegfa with stunted sprouting, reduced branching and increased sprout diameter (*Figure 3c*). Emerald expression was also strongest in clusters of cells at the front (*Figure 3d*). Computational modelling further predicted that

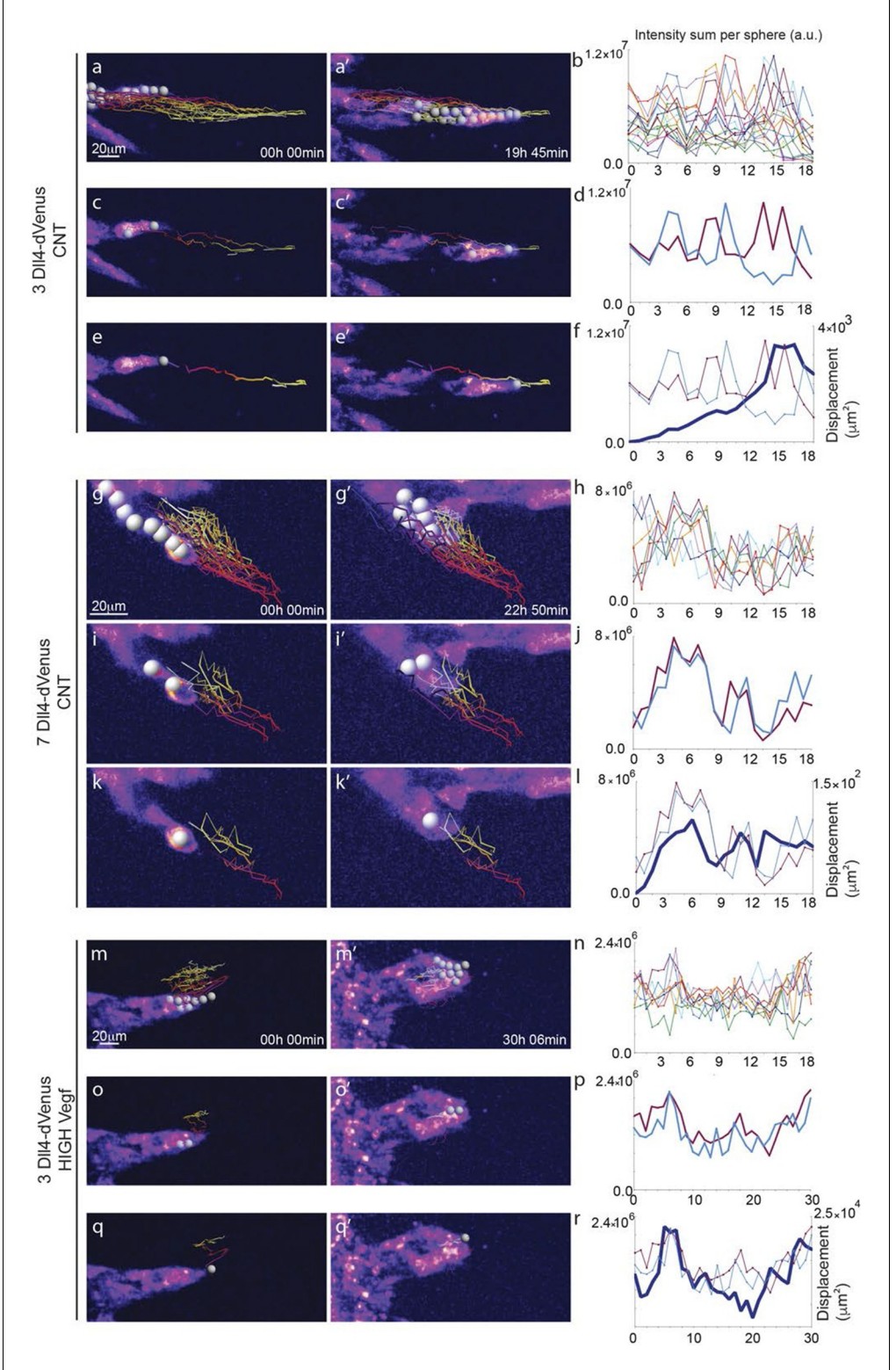

**Figure 4.** Time-lapse analysis of *Dll4* reporter levels identify Dll4/Vegf dose- dependent shift from 'individual cell' to 'synchronous cell' signalling and behavior. Start (**a, c, e, g, i, k, m, o, q**) and end (**a', c', e', g', i', k', m', o', q'**) point of confocal time-lapse acquisitions of 3*Dll4*-dVenus (**a-f**) and 7*Dll4*-dVenus (**g-l**) embryoid bodies cultured in normal Vegf conditions (50 ng/ml), and 3*Dll4*-dVenus embryoid bodies treated with high Vegf concentration (500 ng/ml) (**m-r**). A heatmap intensity range color was used to represent dVenus expression levels. (See supplementary
*Figure 4 continued on next page*

*Figure 4 continued*

information; *Videos 7*, *8*, *9*). (a-a', g-g', m-m') Arbitrary 10 µm spheres were employed to fill the sprout volume, using the Imaris 'Spot' cell tracking tool. (c-c', i-i', o-o') Sprout volumes covered by two neighboring spheres are monitored overtime. (e-e', k-k', q-q') A single sphere placed at the sprout tip is used to monitor the tip advance and retraction. (**b**, **h**, **n**) Quantification by time-lapse microscopy of dVenus signal levels relative to each sphere volume covering the sprout. dVenus intensity sum is represented by arbitrary units (a. u.). For technical reasons absolute intensity sum values are not comparable between experiments. (**d**, **j**, **p**) Quantification by time-lapse microscopy of dVenus signal intensity sum in two neighboring sphere volumes. (**f**, **l**) Quantification by time-lapse microscopy of the sprout tip displacement along the three axes x, y, z is shown together with dVenus intensity sum from (**d**, **j**). (**r**) To counteract the effect of a sprout drift along the x axis on x-y-z displacement (See supplementary information; *Video 9*) only the y-displacement is quantified by time-lapse microscopy and shown together with dVenus intensity sum from (**p**). (For details on the tracking technique see Materials and methods)

The following figure supplements are available for figure 4:

**Figure supplement 1.** *Dll4*-dVenus reporter expression is dynamically and differentially regulated in vitro in individual endothelial cells.

**Figure supplement 2.** Additional examples of dynamic *Dll4*-reporter intensity profiles and sprout tip displacement illustrating a Dll4- and Vegfa-level dependent switch from individual cell to synchronous cell signaling and behaviour.

---

raising Dll4 levels should also lead to synchronized fluctuations (*Figure 3—figure supplement 1*; *Video 3*, *4* and *5*). Indeed, ES cells carrying multiple copies of the *Dll4*-dVenus transgene (7*Dll4*) showed high but dynamic levels of *Dll4* expression, and synchronized in clusters of cells that correlated with stunted sprouting and vessel expansion (*Figure 3 e-i*). DAPT treatment restored branching and elongation under high Vegfa or *Dll4* overexpression conditions (*Figure 3—figure supplement 2*), demonstrating that this synchronization and morphogenesis effect is Notch mediated.

## Individual Dll4 dynamics correspond to functional sprout elongation

To directly observe *Dll4* dynamics, we established time-lapse imaging of EBs (*Figure 4*). Signal intensity heatmap illustrations highlight the rise and fall in dVenus expression in individual cells (*Figure 4—figure supplement 1*; *Video 6*). Tracing the sprout over time, we quantified the intensity of *Dll4*-dVenus in arbitrary spheres filling the volume of the sprouts (*Figure 4—figure supplement 2*). Intensity plots of the individual spheres showed dynamic behaviour, with temporal profiles similar to those observed in monolayers in vitro with RNA analysis (*Figure 1* and *Figure 4*). However, the profiles between the spheres filling the sprouts showed no synchrony between neighbouring cell

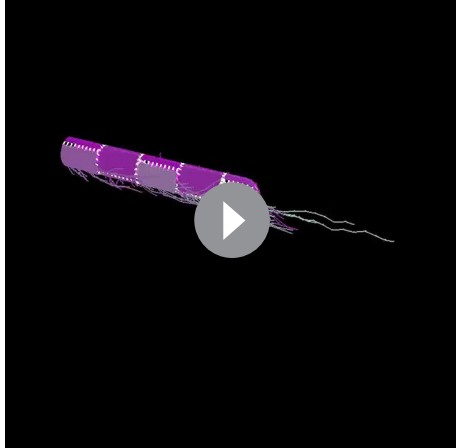

**Video 3.** sprout simulation 2 Dll4 copy (WT).

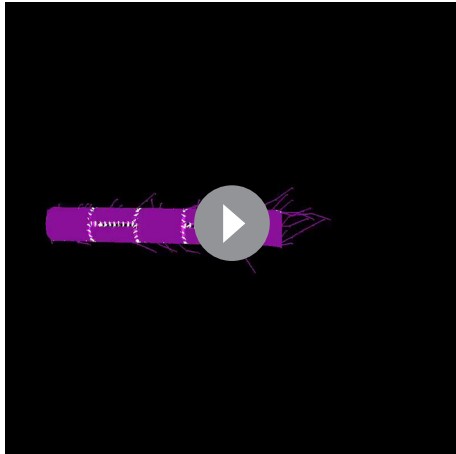

**Video 4.** sprout simulation 3 Dll4 copy.

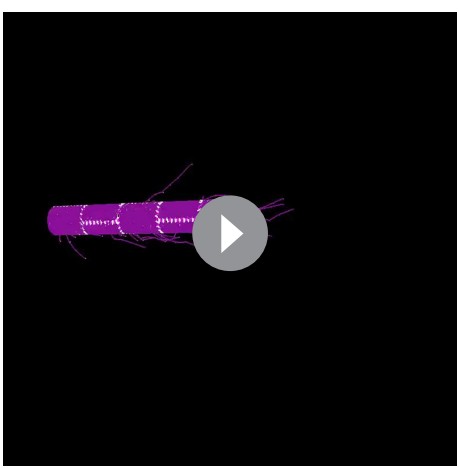

**Video 5.** sprout simulation 7 Dll4 copy.

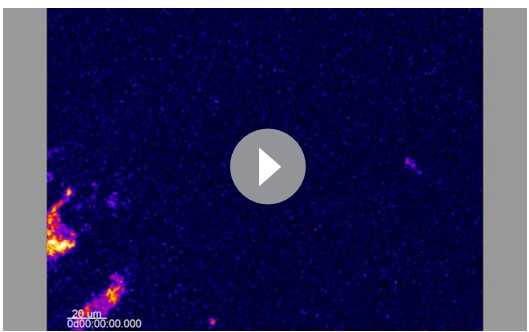

**Video 6.** EB 3*Dll4*-dVenus from *Figure 4* – Related to figure supplement 1.

fluctuations, in line with the salt-and-pepper distribution of Dll4 between neighbouring cells (*Figure 4a–d*). The asynchronous dynamic *Dll4* expression behaviour correlated with a continuous forward movement of the leading tip (*Figure 4e–f*, *Figure 4—figure supplement 2a,b*; *Video 7*, *10* and *11*).

When performing the same dynamic imaging of EBs comprised of ES cells overexpressing *Dll4* (7*Dll4* clone), we observed very similar dynamic fluctuations in individual spheres (*Figure 4g,h*). However, collectively, the patterns of *Dll4* expression between neighboring spheres appeared more synchronized (*Figure 4i,j*). Intriguingly, these wider sprouts moved forward led by a cluster of several cells synchronously expressing high *Dll4*-dVenus levels (*Video 8*, *12* and *13*, *Figure 4—figure supplement 2c,d*). A short time later, the same sprout retracted, concomitant with reduced *Dll4*-dVenus levels in the cluster. Later again, regaining higher expression, the same sprout moved forward with a bright *Dll4*-dVenus cluster at the tip (*Figure 4k,l*). Similarly, high Vegfa levels induced clustering and iterative extension and retraction concomitant with synchronized dll4 reporter fluctuations at the tip (*Figure 4m–r*, *Video 9*, *14* and *15*, *Figure 4—figure supplement 2e,f*). These results provide the first experimental evidence for a synchronized fluctuation in Dll4 expression and cell behaviour by high Vegfa levels or *Dll4* overexpression in an in vitro sprouting assay.

## Dll4 dynamics switch from heterogeneous to synchronised fluctuations in vivo with high Vegfa or high Dll4

To investigate the possibility of synchronized fluctuations in signalling and behaviour in vivo, we injected mVegfa 165 into the vitreous of early post-natal *Dll4* reporter and WT pups (P4) (*Figure 5*). Fluorescent ISH allowed the detection of nascent *dll4* mRNA (one or two dots) in the nucleus as well as mature mRNA in the cytoplasm of actively transcribing ECs (*Figure 5—figure supplement 1*). dVenus and *dll4* mRNA expression showed a salt-and-pepper distribution in control retinas, but strong clustering of *dll4* positive cells in Vegfa injected retinas 20 hr post-injection (*Figure 5a–h*).

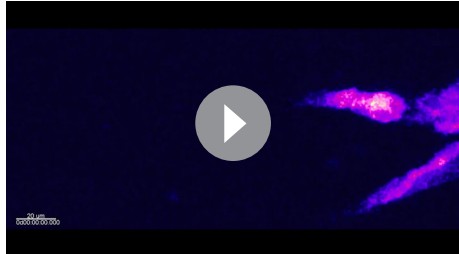

**Video 7.** EB 3*Dll4*-dVenus normal Vegf from *Figure 4*.

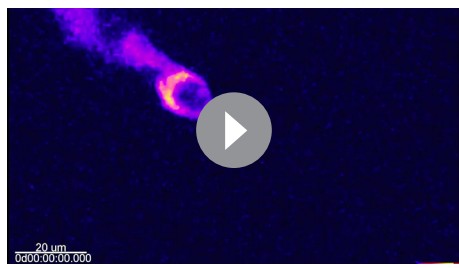

**Video 8.** EB 7*Dll4*-dVenus normal Vegf from *Figure 4*.

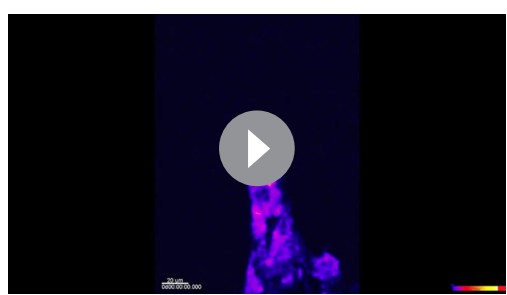

**Video 9.** EB 3*Dll4*-dVenus high Vegf from *Figure 4*.

Intriguingly, despite the strong Vegfa stimulation, clusters of *dll4* negative cells were also present, arguing against a simple linear relationship between Vegf and *Dll4* expression. Similar results were obtained for Dll4 protein expression (*Figure 5—figure supplement 2a–f*). The change in spatial *Dll4* expression distribution correlated with a dramatic shift from branching and extension to vessel enlargement, remarkably similar to what we observed in the EB system (*Figure 3*). At the sprouting front, most of the sprouts had disappeared and were replaced by a dramatically widened peripheral vessel with occasional protrusions.

In the plexus, all vessels increased substantially in diameter while the overall branching frequency was significantly reduced (*Figure 5—figure supplement 3*). Radial expansion of the vascular plexus was also significantly reduced, consistent with earlier reports (*Gerhardt et al., 2003*).

In these experiments Vegf is acutely elevated, by injection, after a normal salt-and-pepper pattern had previously been established through normal development, This scenario is different to the previous simulations where endothelial cells experience high Vegf from the onset, prompting us to re-evaluate the computational predictions in simulations of Vegfa injections at a late timepoint, once the normal salt-and-pepper pattern had established using our existing Vegf/Notch feedback model. We observed that the impact of elevated vegf is strong enough alone to drive a full shift in patterning dynamics matching the observed phenotypic changes in vascular patterning of the retina plexus. Simulations of collagen matrix deposition of advancing sprouts in this model further predicted that the iterative synchronized sprouting and retraction movement generated by the synchronized fluctations in Notch signaling should lead to accumulation of empty collagen sleeves ahead of the vascular front (*Figure 5i*; *Video 16*). Testing this prediction, we assessed the pattern of collagen IV basement membrane deposition at the vascular sprouting front 3 hr (P4 retinas) and 20 hr (P5 retinas) after Vegfa injection. Retinas injected with Vegfa showed a dramatic mismatch between the vasculature and collagen IV deposition in the blunted regions of the sprouting front (*Figure 5j,k*). In these areas, already 3 hr after Vegfa injection, collagen IV sleeves protruded ahead of the blunted vasculature, and further increased in number and length over time (*Figure 5l–o*). The observed radial expansion delay (*Figure 5p*) is thus likely a result of unproductive and iterative sprouting and retraction.

## High Vegfa can drive vessel expansion in the absence of proliferation

A plausible alternative explanation for the observed enlargement in vessel diameter upon high Vegfa levels is cell proliferation. Indeed, injecting Vegfa into the vitreous leads to widespread endothelial proliferation throughout the plexus (*Figure 6a,c*). To discriminate between proliferation and collective endothelial cell synchronization effects, we inhibited proliferation by systemic treatment with mitomycin C (MMC) 24 hr prior to and again concomitant with intraocular mVegfa 165 injection, and analysed proliferation and retinal patterning 24 hr post Vegfa treatment. EdU incorporation revealed a complete block in proliferation in the MMC treated samples (*Figure 6a–d*). Blocking proliferation

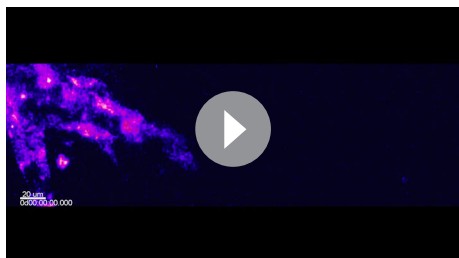

**Video 10.** EB 3*Dll4*-dVenus normal Vegf (*Video 1*) from *Figure 4—figure supplement 2*.

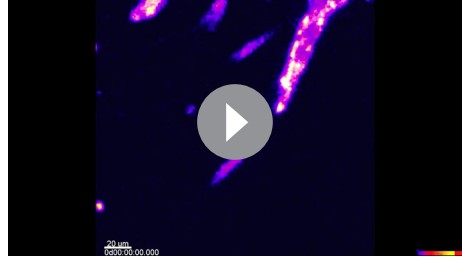

**Video 11.** EB 3*Dll4*-dVenus normal Vegf (*Video 2*) from *Figure 4—figure supplement 2*.

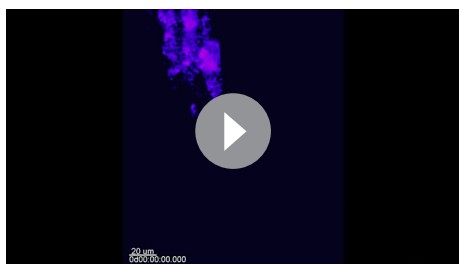

**Video 12.** EB 7*Dll4*-dVenus normal Vegf (*Video 1*) from *Figure 4—figure supplement 2*.

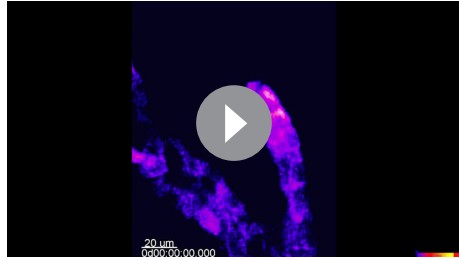

**Video 13.** EB 7*Dll4*-dVenus normal Vegf (*Video 2*) from *Figure 4—figure supplement 2*.

alone without Vegfa treatment lead to reduced vascular density in particular in the peripheral zone that developed over the duration of MMC treatment, in line with the proliferation zone in normal development (*Figure 6b*). Interestingly however, the diameter increase at expense of branching complexity after Vegfa treatment was equally evident in samples from MMC plus Vegfa treated pups (*Figure 6k,l*). To our knowledge, this is the first direct evidence for the morphogenic effects of quantitative changes in Vegfa to be independent of proliferation. Although proliferation will likely contribute to vessel size increase under high Vegfa, surprisingly, it is not required. Instead we propose that the increased coupling of cells through elevated Dll4/Notch signaling, drives clustering of cells through synchronization of cellular dynamics that interfere with branching, but promote vessel diameter increase.

## Synchronised fluctuations of Dll4 in pathological angiogenesis

These observations also suggest that the pathological vascular patterning in disease conditions with elevated Vegfa levels might be a consequence of synchronized *Dll4* dynamics and the consequent abnormal iterative sprouting and retraction behaviour or cell rearrangement abrogation. To directly test this prediction, we investigated the expression and distribution of dVenus, *dll4* mRNA and protein as well as collagen IV empty sleeves formation in two different pathological systems with aberrant vascular patterning; the oxygen induced retinopathy (OIR) model, which reproduces aspects of the pathobiology of human retinopathies (*Smith et al., 1994*) and the glioblastoma brain tumour model (GBM).

In OIR, temporal vessel regression leads to ischemia of the neural retina, inducing a neovascular response that is characterized by dramatic vessel expansion and ineffective ballooning sprouts (glomeruloid tufts) that penetrate the inner limiting membrane (*Smith et al., 1994*). In 3*Dll4*-dVenus reporter mice, OIR leads to strong clustering of endothelial cells positive or negative for dVenus in the neovascular tufts (*Figure 7a–d*). Similar to Vegfa injection, OIR also leads to an overall increase in *Dll4*-dVenus levels (*Figure 7a*). *dll4* mRNA and protein expression in WT retinas confirmed clustering of cells with high and low expression, compatible with the idea of localized synchronization of endothelial *Dll4* fluctuations in OIR (*Figure 7e–h*; *Figure 5—figure supplement 2g–i*).

Collagen IV deposition poorly matched the vasculature, revealing that substantial extensions had previously formed beyond the sprouting front and tufts in the OIR retinas (*Figure 7i–k*). These

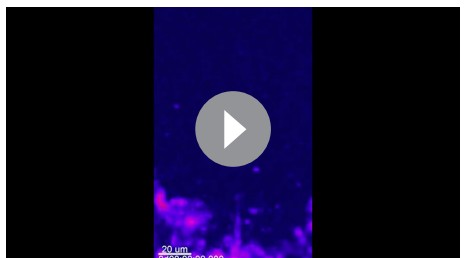

**Video 14.** EB 3*Dll4*-dVenus high Vegf (*Video 1*) from *Figure 4—figure supplement 2*.

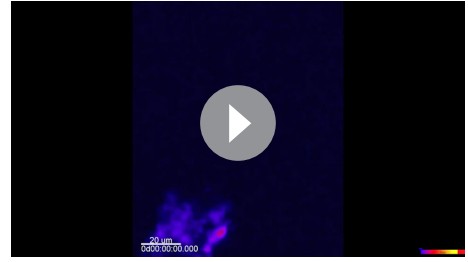

**Video 15.** EB 3*Dll4*-dVenus high Vegf (*Video 2*) from *Figure 4—figure supplement 2*.

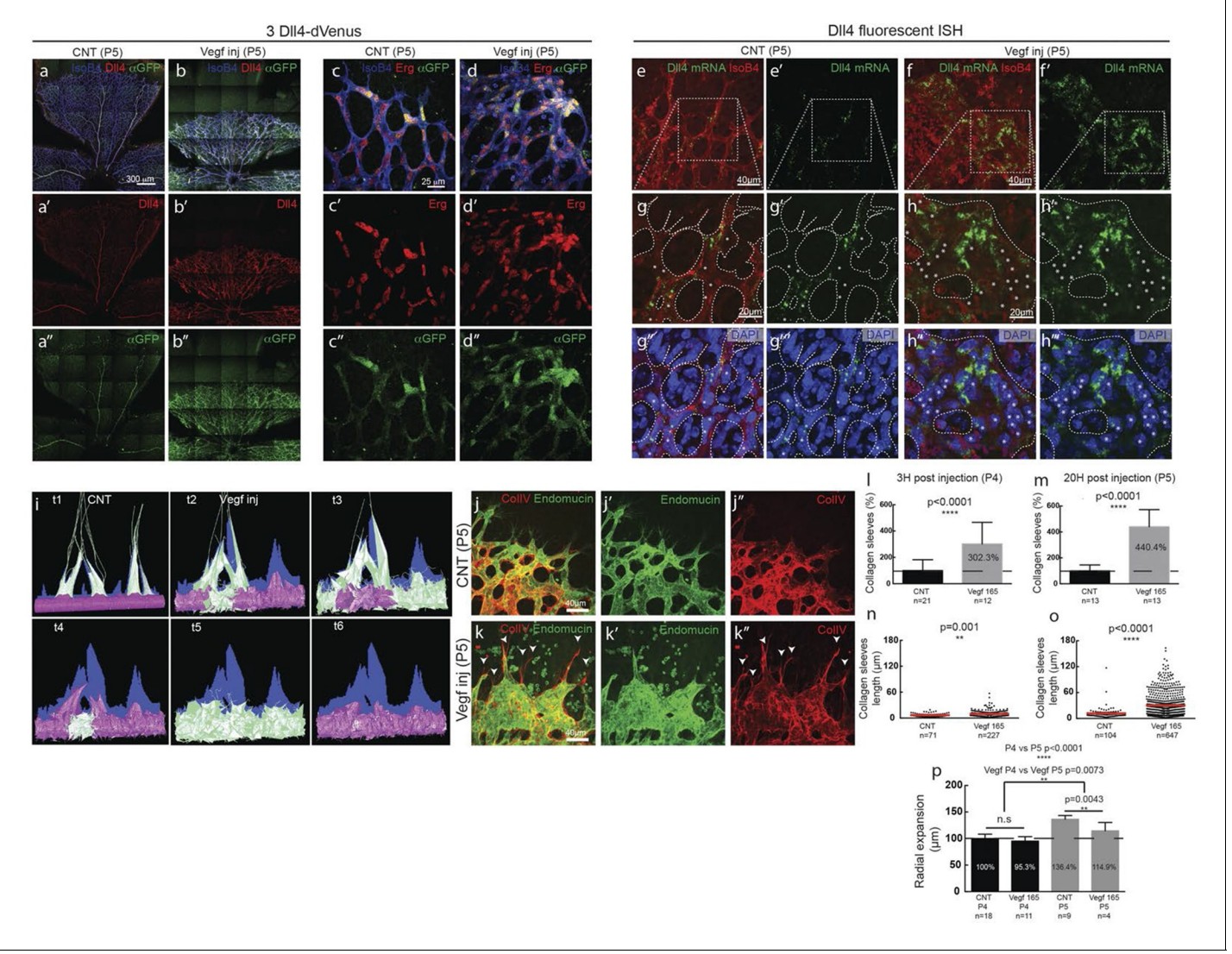

**Figure 5.** Endothelial cell Dll4 expression synchronization under high Vegfa in vivo. (a–d) Dll4 (red in a, b), ERG (red in c, d), anti-GFP (green) and isolectin B4 (blue) staining in representative overview tile-scan (a, b) and high magnification (c, d) images of whole-mounted 3Dll4-dVenus P5 retinas not injected (CNT; a, c) and injected with mVegfa165 (Vegf inj; b, d). For anti-GFP (dVenus) and ERG signal a median filter of 3 and 5 pixel, respectively, was used. (e–h) Representative confocal images of whole-mounted WT P5 retina not injected (e) and mVegfa165 injected (f), labeled for *dll4* mRNA detected by fluorescent ISH (green) and isolectin B4 (red). White dashed boxed areas in each panel (e, f) are magnified in (g,h) images. To facilitate endothelial cell nuclei visualization (DAPI; blue) together with *dll4* mRNA only one stack is shown in panels g and h. White dashed lines delimit endothelial cells (Iso B4). Asterisks represent EC negative for *dll4* mRNA expression. (i) Computational simulation of collagen IV deposition (blue) after high Vegfa stimulation (for details on simulation, see Materials and methods). At t1 a normal Vegfa condition with a linear gradient extending above the sprout is simulated. High uniform Vegfa levels are simulated from t2 through t6. Cells with high Dll4 expression and 'tip cell phenotype' are represented in green; cells with low Dll4 expression and 'stalk cells phenotype' are shown in purple. (j, k) Representative confocal images of the collagen IV distribution at the sprouting front of WT P5 retinas not injected (j) and injected with mVegfa165 (k). Collagen IV is shown in red; Endomucin in green. Arrows indicate empty collagen IV sleeves. (l–o) Quantification of the total number (l, m) and length of empty collagen IV sleeves (n, o) in WT retinas three (l, n) and twenty (m, o) hours post-injection. Mean ± S.D values are indicated (n, o). n= number of retinas analyzed (l, m); n= total number of collagen sleeves observed (n, o). P values calculated using a two-tailed, unpaired t test. (p) Quantification of the radial expansion in P4 (0H post injection) and P5 (20H post injection) WT retinas not injected and injected with mVegfa165. n= number of retinas analyzed. Values represent mean ± S.D. p values calculated using a two-tailed, unpaired t test.

The following figure supplements are available for figure 5:

**Figure supplement 1.** Detection of mature and nascent *dll4* mRNA transcripts using whole mount fluorescent ISH.

**Figure supplement 2.** Dll4 protein expression is synchronized under pathological Vegfa concentration.

*Figure 5 continued on next page*

*Figure 5 continued*

**Figure supplement 3.** High Vegf concentrations during retinal angiogenesis result in aberrant vascular patterning.

results indicate that the glomeruloid tuft formation in OIR is associated with iterative extension and retraction of endothelial processes, which however fail to establish functional new branches.

Angiogenesis is one of the hallmarks of cancer (*Hanahan and Weinberg, 2011*). However, tumour vessels show highly abberant and dysfunctional patterns, with diameter variability and irregular branching. In many tumours, including glioblastoma, hypoxia driven Vegfa expression drives the angiogenic response. To investigate the implications of Dll4/Notch signalling and behavior synchronization between endothelial cells for vascular patterning in tumours, we analyzed *dll4* mRNA and protein expresion together with collagen IV deposition in a mouse syngeneic glioblastoma model (GBM). C57Bl6 derived CT-2A glioblastoma cell spheres implanted into the brain of mice developed highly vascularized solid tumours over the course of 4 weeks (*Martínez-Murillo and Martínez, 2007*). Tumours expressed high levels of Vegfa and Hif1a indicating hypoxia, and developed a highly irregular vasculature (*Figure 8—figure supplement 1*). *dll4* fluorescent ISH performed on 200 μm vibratome sections of GBM revealed a highly irregular pattern of expression (*Figure 8a*). Whilst some vessels showed little or no signal at all, other vessels showed high levels of expression in all neighboring endothelial cells (*Figure 8 a–c*). Vessels close to the hypoxic core generally exhibited the highest expression and strong clustering frequently associated with local vessel diameter increase. Similarly Dll4 protein expression showed clustering of high levels in adjacent cells (*Figure 5—figure supplement 2j,k*). Furthermore, the aberrant vascular patterning in GBM was associated with extensive empty collagen IV sleeves radiating from the tumour vessels (*Figure 8d–f*). These results provide the first evidence for iterative sprouting and retraction and, together with the *Dll4* expression pattern, indicate that endothelial cell Dll4/Notch signalling and sprouting behaviour becomes locally synchronized between neighboring cells during tumour angiogenesis.

Together our results demonstrate that the salt-and-pepper differential pattern of *Dll4* expression normally associated with vascular branching switches to synchronized fluctuations under conditions of experimental high Vegfa, in retinopathy and tumour angiogenesis. Dynamic observations in the EB model suggest that nonlinear synchronized fluctations in Dll4/Notch drive vessel expansion and disrupt effective cell rearrangement and migration leading to reduced branching and elongation.

## Discussion

Our current results provide evidence that Notch signalling activity, between activated endothelial cells, undergoes a phase transition between two distinct operational modes, depending on the levels of Vegfa. At normal physiological levels, that typically correspond with developmental angiogenesis, Dll4 fluctuates dynamically in individual cells; the lateral-inhibition feedback with the Vegfr system functions to generate differences between neighbouring cells that manifest themselves as a salt-and-pepper distribution of *Dll4* high and low cells. As a consequence, the salt-and-pepper distribution of tip and stalk cell phenotypes, although individually transient, support the differential behaviour of neighbouring cells at any given time. Dynamic observations in embryoid body sprouting assays indicate that the differential behaviour is critical for branching and continuous elongation. At higher Vegfa levels however, the dynamic fluctuations between neighbouring cells become synchronized, corresponding with a loss of salt-and-pepper

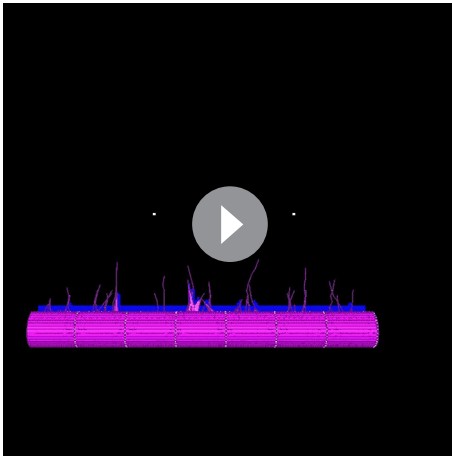

**Video 16.** Collagen IV sleeve simulation.

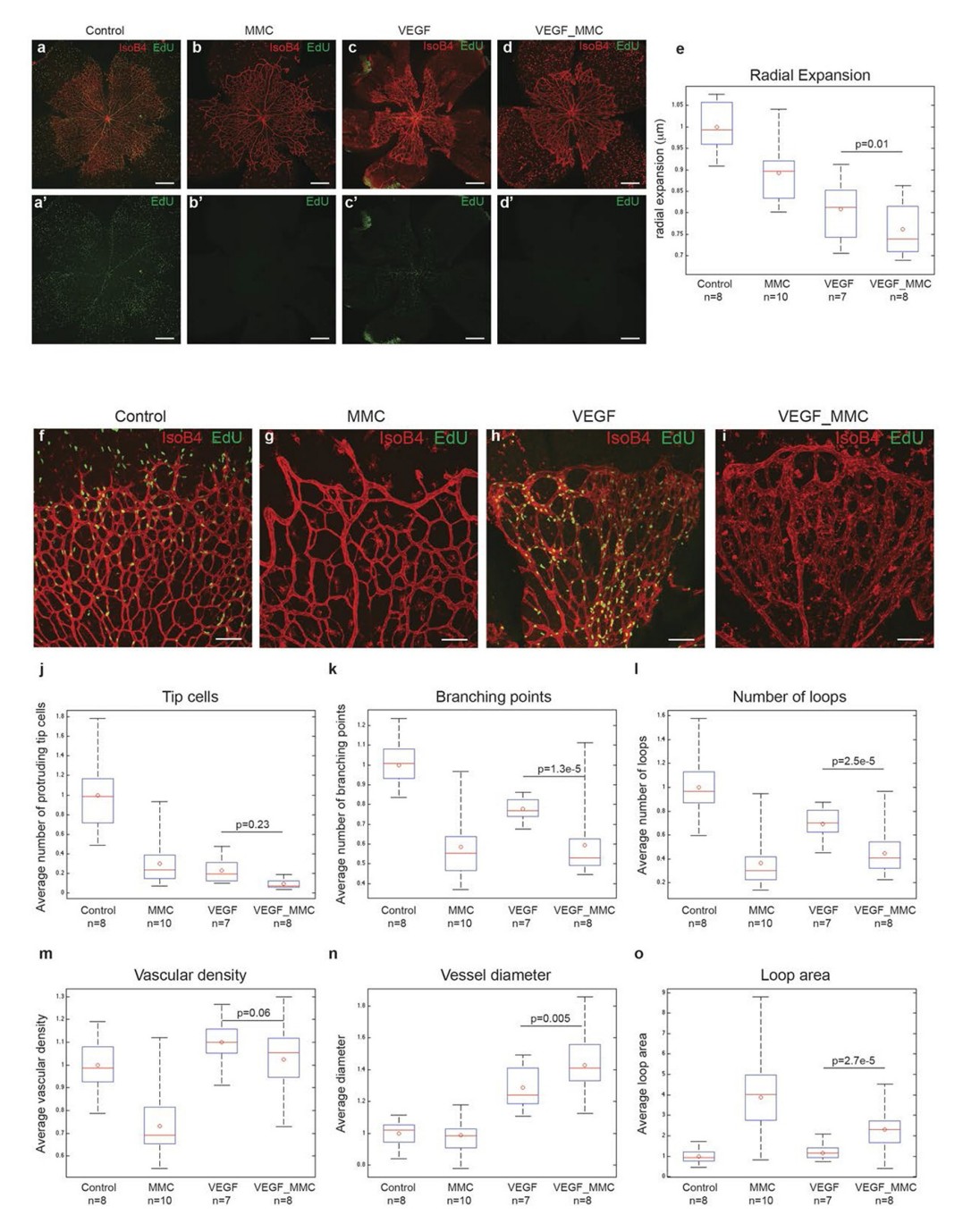

**Figure 6.** High levels of Vegfa induce vessel expansion even in absence of EC proliferation. Representative image of a tile scan showing IsoB4 (red) and EdU staining (green) in non treated (control; a–a'), MMC (bb'), Vegfa 165 (cc') and MMC_Vegfa 165 (d–d') treated retinas. (e) Quantification of the radial expansion in control (a), MMC (b), Vegfa 165 (c) and MMC_Vegfa 165 (d) treated retinas. High magnification images showing IsoB4 (red) and EdU staining (green) in non-treated (control; f), MMC (g), Vegfa 165 (h) and MMC_Vegfa 165 (i) treated retinas. Quantification of the number of tip cells (j), branching points (k), number of loops (l), vascular density (m), vessel diameter (n) and loop area (o) in non treated (control), MMC, Vegfa 165 and MMC_Vegfa 165 treated retinas. Scale bar correspond to 400 µm (a–d') and 100 µm (fi) respectively. Statistical comparison and number of animals analyzed are indicated in the graphs.

patterning. This synchronization, and thus local clustering of cells with high or low Dll4 levels, is dependent on Notch activity, as dampening or inhibiting Notch activity disrupts synchronization. Observations of dynamic behaviour in EBs stimulated with high Vegfa demonstrate that

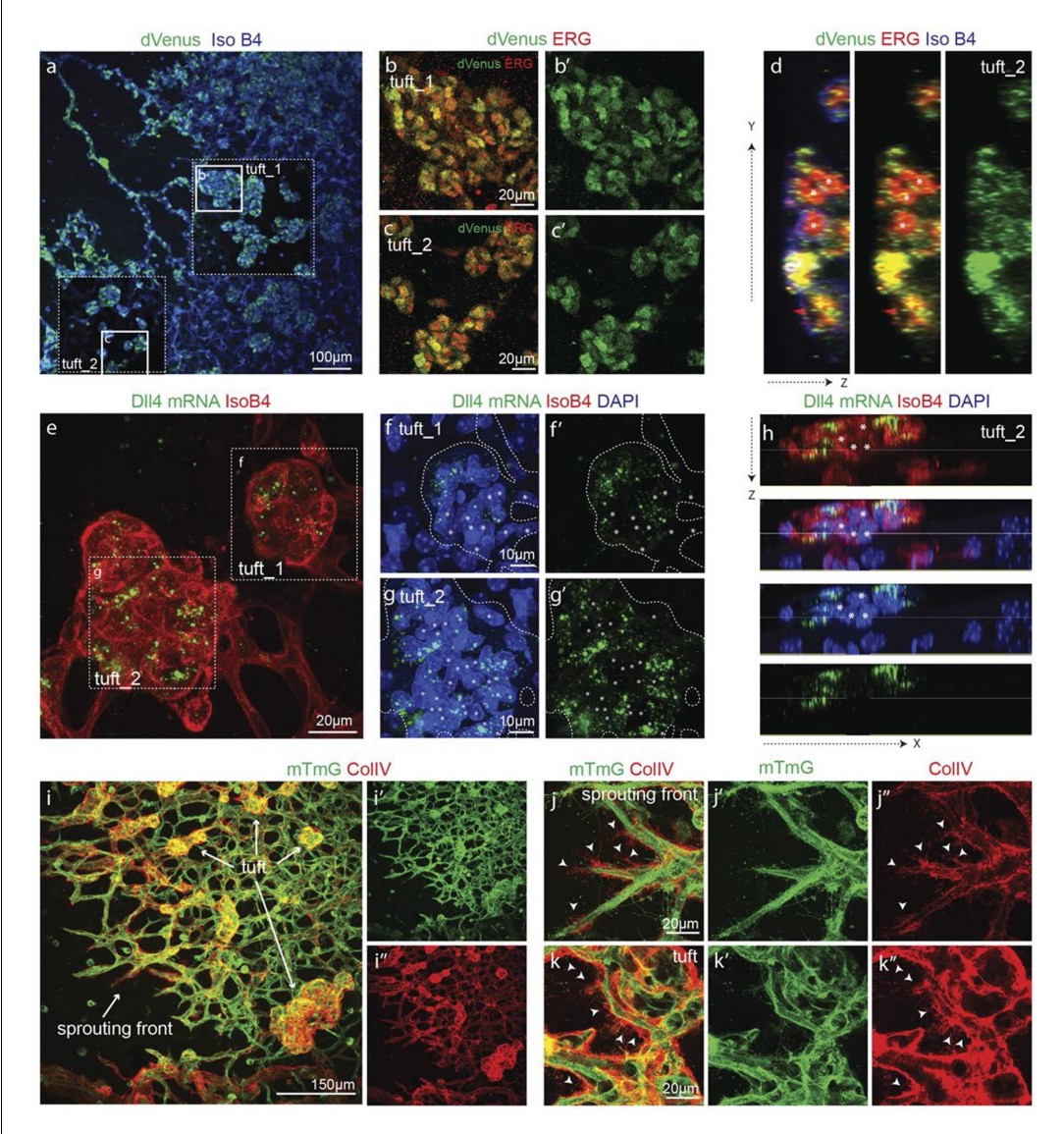

**Figure 7.** Epiretinal tufts in oxygen induced retinopathy show Dll4 expression sychronization. (ac) Representative overview tile-scan of a whole-mounted P15 3*Dll4*-dVenus OIR retina (a). White dashed boxes highlight two different tuft regions; to facilitate tufts visualization only one optical section is shown in the boxed area. Full line boxes are magnified in **b** and **c**. For dVenus and ERG signals (**b**, **c**) a median filter of 3 and 5 pixel, respectively, was used. dVenus (anti-GFP) is shown in green, endothelial nuclei are labeled in red (ERG) and endothelial cells in blue (Iso B4). (**d**) Y–Z confocal section of (**c**). dVenus (anti-GFP) expression is shown in green, endothelial cell are labeled in blue (Iso B4) and endothelial nuclei in red (ERG). Asterisks represent EC negative for dVenus expression. (**e–g**) Representative confocal overview image (**e**) and high magnification images (**f**, **g**) of two different tufts in a WT P15 OIR retina. *dll4* mRNA, detected by fluorescent ISH, is shown in green, endothelial cells (Iso B4) are shown in red and endothelial nuclei in blue (DAPI). White dashed boxes in panel (**e**) highlight the regions of the tufts analyzed in **f** and **g**. In **f** and **g**, white dashed lines delineate endothelial cells on each panel to help visualization. Asterisks represent EC negative for *dll4* mRNA expression. (**h**) X-Z confocal section of (**g**). Stainings for *dll4* mRNA (green), IsoB4 (red) and DAPI (blue) are shown. Asterisks represent EC negative for *dll4* mRNA expression. (**i**) Overview image of collagen IV distribution in a mTmG WT P15 OIR retina, labeled with collagen IV (red). New sprouting front and tuft regions are highlighted in the merged image. (**j**, **k**) Representative images of collagen IV empty sleeves protruding ahead of the new sprouting front (**j**) and of tufts region (**k**) in the mTmG WT p15 OIR retina. Collagen IV is labeled in red. Arrows indicate empty collagen IV sleeves.

synchronization of *Dll4* levels coincides with synchronized sprouting activity. As a consequence, sprouting and branching is disrupted and the vessel instead expands (*Figure 9*). The finding of clustered *Dll4* expression by in situ hybridisation, *Dll4* reporter and protein staining in the vessel

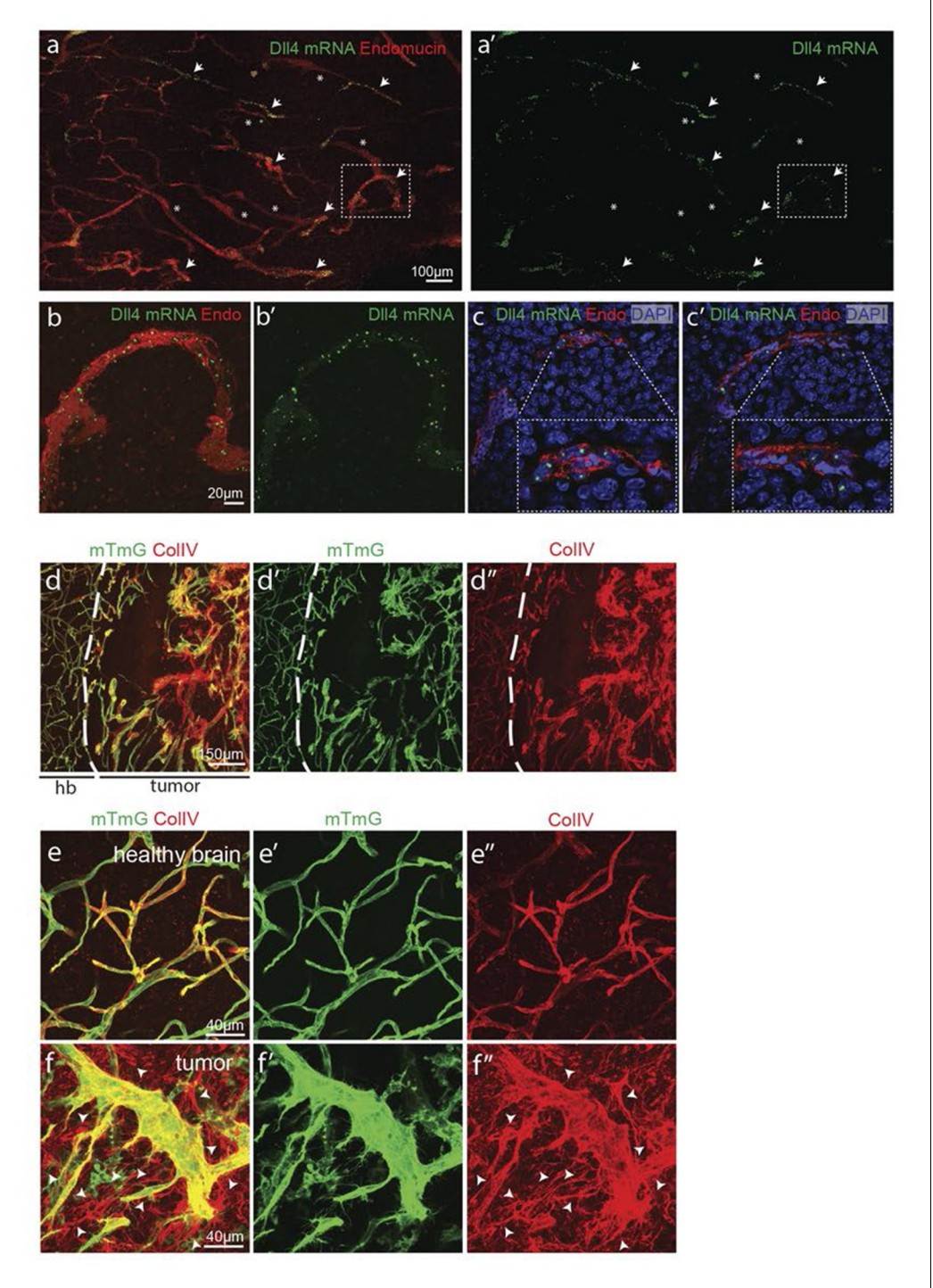

**Figure 8.** Synchronized Dll4 expression and sprouting behavior in tumour angiogenesis. (a) Tile scan representative overview of *dll4* mRNA expression detected using fluorescent ISH (green) in CT-2A glioblastoma tumor vessels labeled with endomucin (red). Asterisks and arrows indicate tumor vessels negative and positive for *dll4* mRNA expression, respectively. White dashed boxes indicate the tumor region analyzed at high magnification on panel b and c. (b, c) High magnification of the positive tumor vessel for *dll4* mRNA highlighted in panel (a). *dll4* mRNA ISH is shown in green, cell nuclei stained with DAPI in blue and endothelial cells, detected using endomucin, in red. To facilitate endothelial cell nuclei visualization together with *dll4* mRNA expression only one stack is shown in the panels where nuclear staining (DAPI; blue) is included (c). (d–f) Confocal overview image (d) and high magnification images (e, f) showing collagen IV (red) deposition around healthy (d, e) and tumor

*Figure 8 continued on next page*

*Figure 8 continued*

vasculature (**d**, **f**) in the glioblastoma tumor model (GBM) developed in a mTmG cre-reporter mouse brain. Endothelial cells express membrane eGFP by *PdgfbiCreERT* tamoxifen-induced recombination. Dashed line separates the healthy brain (hb) from the tumor region on **d**. Arrows indicate empty collagen IV sleeves.

The following figure supplement is available for figure 8:

**Figure supplement 1.** Hypoxic tumor cells induce high Vegfa concentrations leading to vessel expansion in mouse glioblastoma model.

expansions in the OIR model and in dilated vessels after Vegfa injection and in mouse glioblastomas together indicate that synchronization is an important principle of vessel malformation in disease.

*Dll4* overexpression, despite under the normal regulatory control, in the absence of altered Vegfa levels, also leads to the same synchronization and thus clustering and loss of branching. Together with the observed up-regulation of *Dll4* under Vegfa control, this suggests that Vegfa-induced *Dll4* up-regulation is the key to synchronization. The observed clustering, stunted sprouting and expansion upon *Dll4* overexpression in the absence of an altered Vegfa environment further suggests that synchronization itself is the major driver of the switch from branching to expansion. When injecting Vegfa in vivo or raising the concentration in vitro, not only the concentration of Vegfa, but also its spatial distribution changes. We previously proposed that Vegfa injection disrupts the Vegfa gradient, thereby reducing tip cell migration, while driving stalk cell proliferation (*Gerhardt et al., 2003*; *Ruhrberg, 2003*; *Ruhrberg et al., 2002*). Based on static images, we concluded that this shift in the balance between tip migration and stalk proliferation reduces branching while driving expansion. Our new dynamic data, the transient nature of tip and stalk cell phenotypes, and the new insights from blocking proliferation under this condition and from computational modelling give rise to a

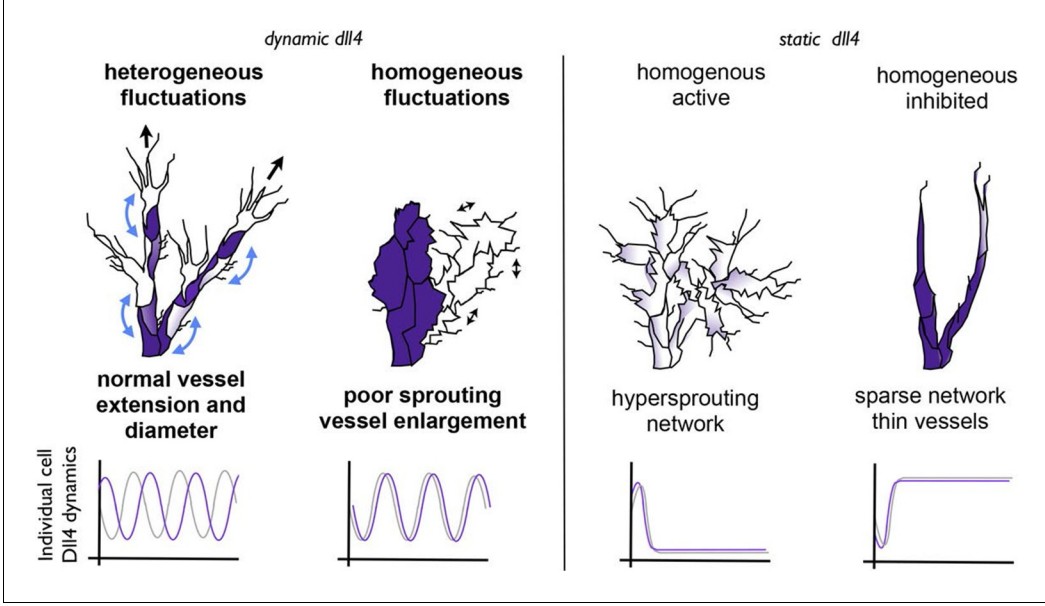

**Figure 9.** Schematic illustrating the important role of dynamic dll4 fluctuations in driving vessel expansion. Individual cells Dll4 levels fluctuate asynchronously during normal vessel growth. Under high VEGF or high Dll4 these fluctuations become more synchronized leading to homogenous cellular dll4 levels that fluctuate between high and low levels together, driving vessel expansion rather than branching. In contrast, when Dll4 levels are homogeneous, but not fluctuating (termed 'static dll4' scenarios here for simplicity) the result is that cells either all remain activated and behave as tip cells (hypersprouting phenotype) or are all constantly inhibited, prohibiting vessel extension and driving a sparse branching phenotype. Thus the expansion phenotype is specifically driven by the dynamic fluctuations between homogeneous Dll4 high/low levels, rather than by the homogeneity of the Dll4 levels alone.

different interpretation. Computational modelling previously predicted that reduced Vegfa gradients interfere with tip/stalk specification as the reduced differences in the local environment make it more difficult for the lateral-inhibition feedback loop to establish differences between neighboring endothelial cells (*Bentley et al., 2008*; *2009*). A loss in gradient already pushes the system towards synchronization, an effect that is dramatically aggravated by higher Vegfa levels. Our present findings indicate that the underlying mechanism driving expansion and clustering, instead of branching and elongation, in situations of reduced Vegfa gradients and high Vegfa levels lies in Dll4/Notch mediated synchronization of adjacent cells as they fail to establish a dynamic salt and pepper pattern.

Intriguingly, overexpression of *Dll4* in tumour cells has been shown to lead to reduced vessel branching and dramatically increased tumour vessel diameter, with evidence that tumour cell Dll4 activates endothelial Notch receptors (*Li et al., 2007*). Why and how this leads to tumour vessel expansion has since remained unclear. In the light of our current results it is tempting to speculate that overexpression of *Dll4* in tumour cells adjacent to the endothelium will cause synchronization of endothelial Notch dynamics, similar to the situation of overexpressed endothelial *Dll4*. Conversely, inhibition of Dll4 by antibody, genetic haploinsufficiency or Notch inhibition by DAPT all lead to a dramatic increase in vessel branching, but reduced vessel diameter in tumour vessels (*Ridgway et al., 2006*; *Noguera-Troise et al., 2006*). We propose that this switch in tumour vessel patterning is driven by the loss of synchronization, and thus a switch from collectively synchronized cell behaviour to differential individual behaviour. Given our recent identification of Notch-regulated differential VE-cadherin adhesion (*Bentley et al., 2014*), it is conceivable that synchronization leads to vessel expansion through the loss of adhesion differences. According to this concept, differential adhesion drives intercalation and thus vessel elongation, whereas synchronization disrupts intercalation, thus causing cells to cluster and thereby enlarge vessel diameter. As endothelial cell migration as opposed to endothelial cell apoptosis appears to present the key mechanism in vessel remodelling (*Franco et al., 2015*), our current finding of Vegfa-induced vessel enlargement and loss of branching in the absence of proliferation further supports the idea that synchronization disrupts the normal differential migration/intercalation behaviour of endothelial cells and thereby leads to the observed morphological changes. Our observations in EBs overexpressing *Dll4* or treated with high Vegfa levels demonstrate that Notch inhibition is sufficient to break the synchronisation and restore branching morphogenesis.

Other developmental morphogenesis systems make use of Notch driven synchronization, a phenomenon first described and best understood in somitogenesis, the formation of mesodermal tissue blocks that form the segmental pattern of vertebrates (*Jiang, 2000*; *Tajbakhsh and Spörle, 1998*). The presomitic mesoderm close to the tip of the tail bud shows dynamic activity of Dll/Notch and Notch target genes of *hes/hey* family. Intriguingly, the zone closest to the tip shows unsynchronized salt-and-pepper patterning of this activity, whilst temporal waves of synchronized activity emerge just rostral of this zone. Although many genes and components of the Notch, Fgf and Wnt-signaling pathway oscillate in the presomitic mesoderm (*Aulehla et al., 2008*; *Wahl et al., 2007*; *Goldbeter and Pourquié, 2008*) and single cells oscillate autonomously through feedback loops in the regulation of the hes family of transcriptional repressors (*Kageyama et al., 2007*; *Webb et al., 2016*), it has become clear that Dll/Notch activity between the cells is the driver of synchronization. In the absence of Notch activity or Dll expression cells gradually drift out of synchrony, leading to loss of somite pattern and thus vertebrae defects. Recent work by Oates and colleagues using mathematical modelling and experimentation identified that the levels of Dll expression not only critically influence synchronization, but also affect the periodicity of the synchronized oscillations by changing the strength of coupling between the neighboring cells (*Herrgen et al., 2010*).

Whether cell-autonomous hes-driven oscillations occur in endothelial cells remains unclear, but speculations on waves of activity of the Bmp and Notch pathway in angiogenesis have recently been raised (*Beets et al., 2013*). Our computational model does not simulate cell-intrinsic oscillations that are coupled via Dll4/Notch. Yet, the ultimate behaviour of the Vegfa-Dll4-Notch-Vegfr feedback loop shows highly similar elements when Dll4 levels change. Thus, although the precise wiring of the signalling circuit is different, there is a striking analogy in that rising Dll4 levels under the influence of Vegfa stimulation will increase the coupling strength between neighbouring endothelial cells. This will ultimately drive the cells to switch from differential to synchronized behaviour and thus to iterative sprouting and retraction, disrupting branching morphogenesis.

Although Vegfa has long been known to have different effects at different concentrations, akin to a morphogen, the underlying mechanism has not been understood. Our current results provide a mechanistic explanation for the dosage effects of Vegfa. Further, the identified switch in behaviour from differential to synchronized by changing Dll4/Notch coupling strength between endothelial cells under the influence of rising Vegfa concentrations identifies a novel, and to date unique, mechanism of action for a morphogen.

From an applied perspective, the identified mechanism has wide reaching implications for our understanding of the effects of anti-Vegfa therapy in cancer, and will hopefully stimulate new research into dosing regimes of both Vegfa and Dll4/Notch inhibitors, separate and in combination, and effects on vessel normalization.

## Materials and methods

### Mice and treatments

Mice were maintained at London Research Institute under standard husbandry conditions. All protocols were approved by the UK Home Office (P80/2391). Glioblastoma studies were performed at the Vesalius Research Center, VIB, KU Leuven where housing and all experimental animal procedures were performed in accordance with Belgian law on animal care and were approved by the Institutional Animal Care and Research Advisory Committee of the K. U. Leuven (P105/2012).

WT animals, the mouse lines Pdgfb-iCre ERT (*Claxton et al., 2008*) and R26mTmG (*Muzumdar et al., 2007*) bred on a C57Bl/6 background were used in this investigation, together with the *Dll4* reporters mouse lines 3*Dll4*-dVenus and 3*Dll4*-Emerald generated for this study. Details of the different mice treatments are specified below.

### Cloning strategy of destabilised and stable Dll4 reporters, ES cells and mouse lines generation

A 500 bps region encompassing the last intron and exon 11 (without the STOP codon) of mouse *Dll4* gene (5' homologous region, 5'HR) was amplified by PCR from a genomic *Dll4* BAC clone (RP23_46P4, BACPAC CHORI) and inserted into the pEnt-Emr/Tet vector, which bears a P2A sequence validated in mouse (*Hsiao et al., 2008*).

For dVenus reporter, destabilised version of Venus (dVenus) coding sequence (CDS) was amplified by PCR and inserted in the same vector, in frame with *Dll4* exon 11 and the P2A sequence. For Emerald reporter, Emerald CDS was already present in the original pEnt-Emr/Tet vector in frame with the P2A sequence.

The first 500 bps of mouse *Dll4* 3' UTR (3' homologous region, 3'HR) were amplified by PCR from the same *Dll4* genomic BAC clone and inserted into the PL45 vector (kindly provided by National Cancer Institute Fredericks), which contains a Neomycin/Kanamycin resistance cassette flanked by two *loxP* sites and driven by a prokaryotic promoter (*em7*) and a eucaryotic promoter (*Pgk*).

*Dll4* 5'HR/P2A/dVenus CDS cassette was released by KpnI and SalI digestion and ligated into PL45/ *Dll4* 3'HR digested with the same enzymes.

The targeting vectors so generated (GenBank accession numbers: BankIt1637803 pTVDll42AdVenus KF293660, *Dll4*-dVenus; BankIt1641166 pTVDll42AEmerald KF293661, *Dll4*-Emerald) were linearized using KpnI restriction enzyme and inserted by homologous recombination (*Yu et al., 2000*) between exon 11 and 3'UTR of the *Dll4* gene contained in the *Dll4* genomic BAC clone, employing SW105 bacteria (kindly provided by National Cancer Institute Fredericks). Chloramphenicol and Neomycin/Kanamycin resistance cassettes were utilised to select clones that have undergone recombination. Recombined *Dll4*-dVenus and *Dll4*-Emerald BAC clones were linearized by AscI digestion, purified and electroporated into mouse ES cells by standard protocols (*Joyner, 2000*) with a Biorad Gene pulser electroporator.

Positive ES cells were selected for Neomycin resistance. Random insertion and homologous recombination of one or two Dll4 alleles were screened with a ViiA7 Real-Time PCR System and the TaqMan Copy Number Assay (Life technologies, Carlsbad, California, see manufacturer for details). Primers for *Dll4* (Mm00537881_cn), dVenus (custom made, AI6RNTP) and Emerald (Mr00660654_cn) were employed together with the Mouse TaqMan Copy Number Reference Assay, Tfrc (Life technologies), to detect the copy number of *Dll4* gene, dVenus and Emerald in the ES genome. ES clones

with single homologous recombination were selected, together with clones bearing single (3 *Dll4*-dVenus and 3 *Dll4*-Emerald) or multiple (7 *Dll4*-dVenus) random integrations. To enable dynamic observation of Dll4 reporter in individual endothelial cells the ES clones selected were used to generate embryoid bodies in 3D matrices.

ES clones with single homologous recombination and with single (3 *Dll4*-dVenus and 3 *Dll4*-Emerald) random integration were also employed to generate mouse reporter lines; briefly, 10–15 ES cells were injected into blastocyst stage embryos collected from C57BL/6J female mice that had been mated to C57BL/6J male mice. Injected embryos were transferred to pseudopregnant recipient (day 2.5 dpc) mice according to standard protocols (*Nagy, 2000*).

Mouse lines carrying double *Dll4*-dVenus homologous recombination were generated by further mice crossing.

Offspring was screened and genotyped by performing the same TaqMan Copy Number Assay (Life technologies) on Dll4 gene, Venus and Emerald used to screen ES cells.

## Intraocular injection

Intraocular injections of 300ng murine Vegfa 165 (Product 450–32; Peprotech, Rocky Hill, NJ), were performed at postnatal day 4 (P4) under isoflurane anaesthesia. Injections were performed using 10 µl gas-tight Hamilton syringes equipped with 34 gauge needles attached to a micromanipulator. Eyes were collected 3H (P4) or 20H (P5) later to proceed with the different experiments. For a detailed description of the experiments in which Vegf injected retinas have been used, together with the conditions required for eye and retina preparation see the specific section of Materials and methods.

## Mitomycin C treatment

WT pups were IP injected with 20 µl/g of a 0.5 mg/ml Mitomycin C (Sigma, St. Louis, Missouri) solution 24 hr (postnatal day 3; P3) and directly before intraocular injection of murine Vegfa 165 at postnatal day 4 (P4) was performed. At P5, 17H after Vegfa injection, all pups were IP injected with 5-ethynyl-2′-deoxyuridine (EdU; Invitrogen, ). Eyes were collected 3H later EdU injection was performed to proceed with retina dissection and EdU_Isolectin B4 staining. (see EdU staining section below)

## Oxygen induce retinopathy model (OIR)

Pathological eye angiogenesis was induced as described in *Smith et al. (1994)*. However, the 75% oxygen condition was applied only during P7-9, before returning to normoxia during P10-15. Conditions used for eye fixation and retina preparation are specified for each technique in which OIR samples have been used. When R26mTmG Pdgfb-iCre ERT mice were used under OIR protocol, tamoxifen (Sigma) was injected intraperitoneally (IP; 20 µl/gr from 4 mg/ml stock solution) at postnatal day 13 (P13); eyes were then collected at P15 to allow visualization of the retinal vasculature.

## Glioblastoma (GB) model

Tumor implantation was performed on WT mice or Pdgfb-iCre-mTmG mice (8–12 weeks) injected intraperitoneally with 100 µg/g of tamoxifen 10 days prior surgery. Craniotomy was realised by drilling a circle in between lamboid, sagittal and coronal sutures of the skull on Ketamine/Xylasine anesthetised mouse. 250–500 µm diameter CT-2A glioblastoma cells spheroid were injected in the cortex and sealed with a glass coverslip cemented on top of the mouse skull. Human end point of the experiment was reached when the tumor exceed 4 mm diameter or if the animal loosed 15–20% of its original weight. Anesthetised mouse was then intracardially perfused with 2% PFA solution. Mouse brain was harvested and fixed overnight in 4% PFA at 4°C. For in situ hybridization, brain was post-fixed in -20°C cold methanol and store at -80°C prior vibratome sectioning. For immunocytochemistry, brain was washes with PBS and sectioned at the vibratome (200 µm thickness sections).

## Cell culture experiments

### bEND5

Mouse brain endothelial cells (bEND5) were cultured on plates pre-coated with 0.2% Gelatin (Sigma), in DMEM (Gibco, ) supplemented with 10% FBS (Gibco) untill confluence was reached. Prior

to Vegf treatment, cells were starved for 4H with serum-depleted medium and then stimulated with either 50 ng/ml (normal) or 1 μg/ml (high) Vegf during 9H or 24H. For 0 Vegf and DAPT conditions, either fresh serum-depleted medium (0 Vegf) or medium supplemented with 1 μg/ml Vegf and 50 μM DAPT were added after cell starvation. Sample collection was performed every hour and Dll4 expression analyzed by quantitative real-time PCR assay.

## 3D embryoid bodies (EB) culture

Embryonic stem cells were cultured and embryoid bodies (EBs) were generated as previously described (*Jakobsson et al., 2006*). Briefly, embryonic stem cells were routinely cultured on a layer of irradiated mouse embryonic fibroblasts (DR4) in the presence of leukaemia inhibitory factor (LIF). For experiments, cells were cultured for two passages without feeders, then trypsinized, depleted of LIF and left in suspension as hanging drops during four days. On day four the formed embryoid bodies were transferred to a polymerized collagen I gel (*Jakobsson et al., 2006*) with addition of 50 ng/ml (normal), 500 ng/ml (high) Vegf (Product 450–32; Peprotech, Rocky Hill, NJ), DMSO or DAPT. Medium with normal, high Vegf, DMSO or DAPT (all at concentrations of 5 μM; Sigma-Aldrich, LY-374973) was changed on day 2 after cell plating and every day thereafter. Sprouted EBs were used to perform live imaging analysis of dVenus expression levels or were fixed and stained for Dll4 and dVenus.

## 2D EB culture

To analyse *dll4* and dVenus mRNA levels in endothelial cells derived from embryonic stem cells by quantitative real-time PCR, EBs were culture in 2D. Briefly, EBs were generated following the same procedure described for 3D cultured EBs; at day 4 EBs were seeded on gelatine-coated plates and treated with 50 ng/ml (normal) Vegfa 165 (Product 450–32; Peprotech, Rocky Hill, NJ) supplemented media, allowing the formation of a peripheral vascular plexus in 2D.

## Quantitative real-time PCR

To monitor Dll4 and dVenus gene expression on bEND5 cells and 2D cultured EBs by quantitative real-time PCR analysis samples were collected directly in RLT lysis buffer (RNeasy MicroKit, Qiagen, Germany) and further processed for RNA isolation. Reverse transcription of mRNA was performed using Superscript III reverse transcriptase (Invitrogen) following the manufacturer recommended protocol. Quantitative real-time PCR was performed using a ViiA7 Real-Time PCR System and Taqman gene expression probes for Dll4 (Mm00444619; Applied Biosystems, Foster City, California) and dVenus (AI6RNTP; Applied Biosystems). GAPDH was used as endogenous control to normalize Dll4 and dVenus gene expression.

## Immunofluorescence

### Retina and glioblastoma samples

#### Fixation conditions

Eyes were collected either at P5 or P15 and fixed with 4% PFA in PBS for 2H (Dll4-ERG staining) or 4-5H (Collagen IV-Endomucin) at 4°C; thereafter retinas were dissected in PBS. When IF required dVenus detection pups were perfused and fixed. Briefly, P5 or P15 pups were anaesthetized by IP injection (20 μl/g) of a Ketaset/Hypnovel mix (0.5 mg/ml) and perfused, via left ventricle intracardiac puncture, with PBS first and then by 2% PFA solution. Eyes were collected and fixed in 2% PFA for 2H at 4°C prior to retina dissection in PBS.

For Dll4-ERG staining (not including dVenus detection), blocking and permeabilisation was performed during 2H at RT using a Buffer consisting of 1% FBS (Gibco), 1% BSA (Sigma), 0.5% Triton X100 (Sigma), 0.01% Na-deoxycholate (Sigma) and 0.02% Na Azide (Sigma) in PBS pH 7.4. Primary Dll4 and ERG antibodies at 1:50 and 1:100 dilution, respectively, and the adequate fluorescent secondary antibody (1:200–1:400), diluted in 1:1 Buffer:PBS, were incubated O/N at 4°C in a rocking platform. For Dll4, ERG and dVenus detection, GFP (dVenus) and Dll4 immuno-staining were followed by donkey anti-rabbit Fab fragments incubation (1:100, Jackson's Laboratories), 4% PFA fixation and, finally, ERG immuno-labelling, in order to avoid intensive cross-reaction between anti-GFP and anti-ERG antibodies (both raised in rabbit). See the end of this section for detailed information about the primary and secondary antibodies used in this study.

For Collagen IV or Endomucin staining, blocking and permeabilisation were performed during 2H at RT using 1%BSA-0.5%Triton X100 diluted in PBS pH 7.4. Primary Collagen IV and Endomucin antibodies at 1:400 and 1:50 dilution, respectively, and the adequate fluorescent secondary antibody (1:400) in 0.5%BSA-0.25% Triton X100 were incubated O/N at 4°C in a rocking platform. An equivalent protocol was used to detect Collagen IV on glioblastoma samples. Primary and secondary antibodies are listed at the end of this section.

For Hif-1a staining on glioblastoma samples, vibratome brain slices were blocked and permeabilized in TNBT (0.1M Tris pH 7.4; NaCl 150 mM; 0.5% blocking reagent Perkin Elmer, 0.2% Triton X-100) for 2H at RT. Tissues were incubated with primary antibody Hif-1a (1:100) diluted in TNBT overnight at 4C, washed in TNT (0.1M Tris pH 7.4; NaCl 150 mM; 0.2% Triton X-100) and incubated with appropriate secondary antibody (1:200) diluted in TNBT O/N at 4C. Tissues were washed in TNT and mounted on slides in fluorescent mounting medium (Dako). Primary and secondary antibodies are listed at the end of this section.

## EdU staining

Eyes were collected at P5 (3 hr after EdU injection) and fixed in 2%PFA at 4°C for 4H, subsequently retinas were dissected in PBS. For EdU detection the Click-iT EdU Alexa Fluor 488 Imaging Kit was used (C10337; Invitrogen). Dissected Retinas were transferred to PBS-0.5% Triton at RT for 2H and additionally washed with PBS for 4 times 10 min. After removing the PBS, 100 µl of Click-iT reaction cocktail (following the manufacture indications from Invitrogen) was added to each retina and incubated ON at 4°C. The day after, retinas were washed 3 times with PBS-0.1% Triton during 15 min each and fixed with 4% PFA at RT before to proceed with the Isolectin staining.

When co-staining with isolectin (Iso B4) was required to detect endothelial cells on retinas, samples were equilibrated for 1–2H using PBlec buffer (PBS pH6.8, 1% Triton X-100, 0.1 mM CaCl$_2$, 0.1 mM MgCl$_2$, 0.1 mM MnCl$_2$) and incubated with Iso B4 -488, -568, -647 or -594 pre-labelled (1:200–1:500) O/N at 4°C in a rocking platform.

In all IFs DAPI (Sigma) was used for nuclei labeling. In general, retinas and glioblastoma samples were mounted on slides using Vectashield mounting medium (Vector Labs, H-1000) except when other specifications are given.

Primary antibodies used: Dll4 (R&D Systems, Abingdon, United Kingdom), ERG 1/2/3 (Santa Cruz antibodies, Dallas, Texas), GFP (Abcam, Cambridge, United Kingdom), Collagen IV (AbD Serotec, United Kingdom), Endomucin (Santa Cruz antibodies), Hif-1a (Upstate, Billerica, Massachusetts) and Iso B4 -488, -568, -647 or -594 pre-labelled (Invitrogen).

Secondary antibodies: the adequate Alexa -488, -555, -568 or -647 conjugated (Invitrogen).

### EBs

Dll4 and dVenus (GFP) expression in EBs was detected following the same protocol described for retinas IFs (see above).

## Immunoblotting

Glioblastoma implanted mouse brains were harvested, tumors were dissected out together with the corresponding contralateral hemisphere region and were separately frozen at -80°C. Samples were lysed in RIPA (20 mM Tris pH 7.5, 60 mM NaCl, 1% Triton X-100, 0.5% deoxycholic acid, 0.1% sodium dodecyl sulfate, 10% glycerol, 25 mM ß-glycerophosphate, 50 mM sodium fluoride, 2 mM sodium pyrophosphate, 1 mM sodium orthovanadate, and protease inhibitor cocktail, Calbiochem, Billerica, Massachusetts), homogenized and sonicated. Proteins were quantitated with BCA kit (Pierce, Rockford, IL). Proteins (100 µg) were analyzed by western blotting using anti-Vegfa (0.5 µg/ml, R&D Systems) and anti-ß-actin (0.2 µg/ml, Sigma) antibodies. Membranes were incubated with peroxidase-conjugated secondary antibodies (1:5000; Pierce) for 2H at RT, and proteins were visualized with ECL detection reagents (Pierce) using ImageQuant LAS-4000 (GE Healthcare, United Kingdom) imaging system.

Western blot quantifications of 4CT-2A glioblastoma implanted mice samples were performed using BioRad QuantityOne software. Data are expressed in fold change with healthy brain region as a reference.

### Dll4 fluorescent ISH

in vitro was kindly provided by Dr. Mailhos (*Mailhos et al., 2001*).

Eyes were collected at postnatal day 5 (P5) or 15 (P15), fixed with 4% PFA for 4 hr and immediately dissected in PBS. Retinas were fixed O/N with fresh 4% PFA at 4°C and transferred to methanol (METOH) the day after. Retinas were store at -80°C until *dll4* fluorescent ISH was performed. Glioblastoma samples were fixed overnight in 4% PFA at 4C, post-fixed in -20°C cold METOH, vibratome sectioned (200 μm) and used to perform *dll4* fluorescent ISH as described below.

To develop *dll4* fluorescent ISH, samples were rehydrated through washing steps with 75%, 50% and 25% METOH-PBS-0.1% Tween-20 at RT, treated with Proteinase K (Invitrogen), fixed with 4% PFA-0.1%Glutaraldehyde (Sigma) and pre-hybridized at 65°C for 2 hr with Hb4 pre-hybridization buffer [25% Formamide deionized (Sigma), 25% 20X SCC pH 7, 5 μg/ml yeast RNA (Sigma), 50 μg/ml Heparin (Sigma) and 0.1%Tween-20]. Probe-hybridization was performed O/N at 65°C using Hb4D5 buffer containing 50 μg/ml Dextran Sulphate powder (Sigma) diluted in Hb4 buffer. *dll4* antisense DIG-probe was used at a concentration of 1μ g/ml to prepare the hybridization mix. The following day, samples were thoroughly washed using 50% Formamide deionized (Sigma) in 2XSSCT, 2XSSCT and 0.2XSSCT at 65°C, cooled down in PBS-0.1%Tween and blocked with PBS-0.1%Tween-8% Sheep serum (Sigma) for 2 hr at RT. After blocking, samples were incubated O/N at 4°C with anti-DIG POD antibody (1:500; Roche, Mannheim, Germany). Then samples were washed 4 times 30 min with TBS-0.1%Tween and incubated for 20-30min with fluorescein tyramide solution (TSA; 1:500), diluted on PBS-0.1%Tween, at RT. Subsequently, 0.001% $H_2O_2$ was added to the TSA solution to activate the TSA reaction. After 30–45 min, the TSA reaction was stopped, and the retinas quickly washed with PBS several times. For TSA solution synthesis and products references see: http://wiki.xenbase.org/xenwiki/index.php/Flourescin_Tyramide_Synthesis.

Before proceeding with IsoB4 staining, retinas were washed during 2 days at 4°C in slow agitation with PBS-0.1%Tween, to reduce ISH unspecific signal, and post-fixed with 4%PFA. For tumor brain samples vasculature was stained using endomucin. The specific protocol for IsoB4 and endomucin staining is listed in the immunofluorescence section. In both samples, additionally DAPI staining was used to visualize endothelial cell nuclei. Retinas and tumor brain samples were mounted on slides using Vectashield mounting medium (Vector Labs, Burlingame , California H-1000) and imaged by confocal microscopy.

### Scanning microscopy

Confocal laser scanning microscopy was performed using Carl Zeiss(Germany) LSM710, Carl Zeiss LSM780 and Leica (Germany) TCS SP8 confocal microscope. Images were processed using Imaris (Bitplane), ImageJ, and Adobe Photoshop software.

### EBs time-lapse laser scanning microscopy

For time-lapse microscopy embryoid bodies were cultured in collagen I in glass bottom 24-well plates (MatTek) using a phenol-red free medium to minimise the autofluorescence background (GIBCO). On day six the plate was transferred to a LSM780 laser-scanning microscope (Zeiss; equipped with a motorized stage, incubator S-M and POC-R cultivation system) maintained at 37°C and 5% $CO_2$ with a humidifier. Z-slices were acquired (40–60 per field every 30 min using 2% laser capacity) with a 20x, numerical aperture (NA) 1.0, water-immersion and coverslip corrected dipping objective (Zeiss).

In order to isolate dVenus and Emerald signals from the autofluorescence background before each experiment dVenus and Emerald spectra were identified and recorded in Lambda mode and the time-lapse acquisitions were performed in Online Fingerprinting mode using the previously recorded spectra.

### Quantification measurements and statistical analysis

#### 'Sphere' analysis in confocal EB time-lapse acquisition

Given the impossibility of automatically or manually tracking single cells in most of the time-lapse acquisitions of embryoid bodies, the Imaris 'Spot' cell tracking tool was employed to fill sprout volumes with arbitrary 10 μm spheres.

By covering the sprout with spheres and then by collecting data relative to each of them, the overall status of the sprout together with the differential conditions within the single volumes could be equally monitored every time point of the acquisition.

dVenus expression levels were followed over time by quantifying the Intensity Sum (measured with arbitrary units) of the reporter signal within each sphere.

To highlight the presence of single cell versus synchronous cell signalling and behaviour within the sprouts two neighbouring spheres, supposedly placed on adjacent cells, were arbitrarily chosen and the relative dVenus signal Intensity Sum was calculated trough-out the time-lapse acquisition.

Sprout tip advance and retraction was measured by quantifying the displacement (distance between first and last track point, $\mu m^2$) along the x, y, z axes of a single sphere placed at the edge of the sprout tip. To counteract the effect of a sprout drift along the x axis on the x-y-z displacement (See Supplementary Information, *Video 9*) only the y-displacement was measured in *Figure 4 (q-r)*.

### 'Confocal' quantification measurements and statistical analysis

Sprout density was measured manually counting the total number of sprouts within a 1500x1500 microns area. At least 3 different areas per EB were analyzed.

Sprout diameter corresponds to the maximum width detected considering the first 100 microns from the sprouting tip. Measurements were performed using Image J software. Between 53 and 136 sprouts per condition were analyzed.

Number of nuclei per sprout was measured counting the total number of nuclei within 100 microns from the sprout tip. DAPI was used to visualize cell nuclei. Between 27 to 63 sprouts per condition were included in the quantification.

For all the three parameters quantified, values were relativized to WT means, considered as 100%.

Radial expansion corresponds to the mean distance from the optic nerve to the edge of the sprouting blood vessels. At least 4 measurements per retina were done, and averaged, using Image J software.

Vessel density corresponds to the vascular area, measured by thresholding isolectin B4 signal in ImageJ, divided by the total area of vascularized tissue. At least 4 images of regions between artery and vein were used per retina.

The number of branching points was measured by manually quantifying the branching points of at least 4 images of regions between artery and vein per retina and dividing by the total area of vascularized tissue.

Protruding tip cells correspond with the number of sprouts detected in the retinal sprouting front. At least 4 images of regions between artery and vein were used per retina.

Empty collagen sleeves correspond with collagen IV segments protruding ahead of the sprouting front negative for IsoB4 staining. At least 10 images per retina and condition were analyzed. Sleeve length was determined using ImageJ software.

For all the vascular parameters quantified, data from 3*Dll4*-dVenus/Emerald and Vegf injected retinas were normalized to WT or not injected retinas, respectively, the means of which are expressed as 100%.

Statistical analysis was performed with Prism 6 software (GraphPad) using two-tailed, unpaired t-test.

Quantification of vessel diameter on glioblastoma samples was performed with ImageJ on images from 8 CT-2A glioblastoma implanted mice. Data are expressed in fold change relative to healthy brain control.

## Computational modelling methods

Simulations were developed using the established memAgent-Spring Model (MSM) of Dll4-Notch signalling during sprouting angiogenesis. Cells in this model are comprised of computational agents, which represent sections of the membrane ('memAgents') connected by springs, which confer tension to the cortex beneath the membrane. The cells are initialised in three dimensional space filled with Vegf (using both a gridded lattice and continuous space mappings), memAgents then interact with their local environment and activate their Vegfr-2 and Notch receptors if any Vegf or Dll4 respectively are present leading to dynamic Notch regulated growth of filopodia and cell migration.

## Monolayer simulation

We reduced the resolution of the established MSM to make the model more coarse and speed up calculation time, allowing the simulation of a larger number of cells as needed in a monolayer simulation (a lattice site now represented 1 micron cubic volume instead of 0.5 microns). Each cell was square with sides of 10 microns (comprising 10x10 memAgents in a 2D square agent spring mesh). Each cell was linked by junction springs to its nine square neighbours creating a fully connected checkerboard sheet of simulated endothelial cells. The coarser lattice resulted in the following recalibrated parameters:

Filopodia grow one grid site per timestep (previously this resulted in one timestep representing 15 s to grow one grid site of 0.5 microns, to match measured filopodia dynamics in in vivo zebrafish data in *Bentley et al. [2008]*). Here filopodia extend 1 micron per timestep so we must define one timestep in a coarse grid as representing thirty seconds of sprouting.

In the original MSM model, the Vegfr-2-Dll4-Notch signalling time delays were set so that the lateral inhibition pathway's periodicity matched the thirty minute periodicity of the zebrafish segmentation clock (*Giudicelli and Lewis, 2004*). Here we are matching mouse cell data and so a longer period of 4 hrs was used. The phase behaviour of the system has been established as independent of the precise periodicity, only the time until selection is affected (*Bentley et al., 2008*).

Receptor and ligand maximal levels and the maximum actin level available for filopodia extension, $V_{max}, D_{max}, N, F_{tot}$ as defined in (*Bentley et al., 2008*) were reduced to a quarter of their original level due to the corresponding reduction in cell surface area.

## Modelling 'normal Vegf' in a monolayer

The precise Vegf microenvironment that the cells experience in vitro is not known, but assumed to have no gradient with limited room for cells to extend processes and filopodia in the intercellular space or over neighbouring cells, however short filopodia and cell shape changes do occur and we assume some heterogeneity in Notch signalling is generated.

A completely uniform level of Vegf cannot produce heterogeneity in cells Notch signalling without greater room for the stochastic filopodia growth to establish differences and a prolonged period of time (*Bentley et al., 2008*). Thus with constrained room for filopodia to grow (10 microns of space only) abstracted to above the sheet for simplicity (rather than simulating processes between cells) a shallow gradient was required to obtain at least transient heterogeneity as the cells compete over time.

Vegf was calculated as follows for each grid lattice site g, where Vegf was set to 0.9 for normal Vegf (representing 50 ng) and 18 for high Vegf (1 mg), $z_g$ is the z axis coordinate position of the grid site and G is the define gradient increase, set to 0.1.

$$VEGF_g = VEGF + z_g G$$

The values for G and Vegf under normal conditions were calibrated to generate maximise the time spent in a heterogeneous pattern. Though uniform Vegf was also simulated.

## Blind ended vessel setup for Dll4 copies simulation

Using the coarse model as defined above for the monolayer we now initialised ten cells back around a single tube with radius 3 microns, using a gridded lattice of dimensions $X_g, Y_g, Z_g$ = 90,86,8. The vessel is placed as sprouting from midway up the y axis sprouting in the direction as the x axis the increases, periodic boundary conditions on the sprout are switched off so edge effects at either end of the sprout may take effect. To close off one end of the cylinder all memAgents lying on the edge of the exposed cylinder at the tip end were placed at the same x,y position but with their z coordinate = 1, thus zipped shut adhered to the collagen layer beneath the vessel. All memAgents other than those in filopodia do not move for these simulations to maintain the original vessel shape, without initiation of new branching or complex Vegf environment simulation, lumen formation etc to study the Dll4 dynamics in a single sprout over time in a parsimonious manner.

## Modelling 'normal Vegf' for in vitro sprouting vessels

Again little is known about the specific local gradients and concentrations of Vegf that vessel in a dish experience. A simple static Vegf environment was used. It was assumed that a shallow gradient

would exist with Vegf increasing towards the tip of the sprout. Sprouts normally migrate along collagen fibres in the embryoid body assay, and astrocytes in the retina, so we assume that there is a fibre/astrocyte beneath the sprout with most of the Vegf adhered to it, creating the strongest gradient below the sprout. We also assume there is a low level Vegf gradient felt all around the sprout given the 3D nature of the environment. The Vegf level in each grid site $g$ (with coordinate position $x_g, y_g, z_g$ on the integer grid) is defined as:

$$VEGF_g = \{ \begin{array}{l} V_{const} + x_g V_{col}, if z_g = 1 \\ V_{const} + x_g V_{col} V_{free}, otherwise \end{array}$$

Here $V_{const}, V_{col}, V_{free}$ were set to 2.1, 0.008 and 0.01 respectively and calibrated to these values to exactly match receptor activation dynamics over time as generated by the validated, less coarse model in *Bentley et al. (2009)*.

## Dll4 3 and 7 copies simulation

Normally in the model Dll4 is set to rise with activity of the Vegfr-2 receptor by twofold (the delta parameter from *Bentley et al. [2008]*). Given there are two copies of Dll4 normally this meant that to simulate three and seven copies we simply changed this parameter Delta to 3 and 7 respectively.

## Collagen Sleeves and high Vegf injection simulation

The original tip cell selection model and parameter set was used as in *Bentley et al. (2009)* in order to see the effects of more detailed filopodia and cell shape changes than in the coarse grained model defined above. Cells are initialised around a pre-existing vascular tube, with periodic boundary conditions, two cells per cross section. New tip cells can be selected from it by stimulation with Vegf, which is adhered to a square lattice of collagen extending above the vessel in the y axis (see *Bentley et al. [2009]* for more details).

The following extensions were made to this model here: 1) Collagen is deposited by the cell by creating a blue square in every lattice site that the cell body (excluding filopodia) touches. Collagen here does not currently change the Vegf signalling but merely shows where the cell body has been located. 2) The cell body was allowed to randomly move forward along filopodia tracks with a small probability (0.001 per filopodia/per timestep) whereas previously only contact with another tip cell's filopodia would trigger this migratory behaviour (see *Bentley et al. [2009]* for more details). This was added to improve realism to the model such that tip cells can migrate slowly before they meet another tip cell when this behaviour would then be enhanced.

## Simulating high Vegf injection

After the salt and pepper pattern has been established and cells are close to anastamosing (representing a normal developing retinal vasculature) an injection of high Vegf (100x the normal level) was simulated which was also assumed to eradicate any gradient of Vegf above the vessel.

## Acknowledgements

This work was supported by Cancer Research UK, the Lister Institute of Preventive Medicine, the Leducq Transatlantic Network ARTEMIS, and an ERC starting grant Reshape (311719). KB is also funded by BIDMC, RB is supported by a HFSP long-term fellowship, TM is supported by an EMBO long-term fellowship, and IG, FS, TM and HG were supported by Stichting Tegen Kanker project grant 2012-181. We are grateful to Rosy Manser (Zeiss), Daniel Zicha (Light microscopy), and Ian Rosewell (Transgenic Services) for technical assistance. We thank Julian Lewis -inspiring colleague and mentor in fond memory - and David Ish-Horowitz for discussions, and Erik Sahai for comments on the manuscript.

# Additional information

## Funding

| Funder | Grant reference number | Author |
|---|---|---|
| Cancer Research UK | Vascular Biology Laboratory | Benedetta Ubezio<br>Raquel Agudo Blanco<br>Martin L Jones<br>Anan Ragab<br>Katie Bentley<br>Holger Gerhardt |
| Human Frontier Science Program | Post-doc Fellowship | Raquel Agudo Blanco |
| European Research Council | REshape 311719 | Ilse Geudens<br>Fabio Stanchi<br>Thomas Mathivet<br>Anan Ragab<br>Holger Gerhardt |
| Vlaams Instituut voor Biotechnologie | Vesalius Research Center VIB3 | Ilse Geudens<br>Fabio Stanchi<br>Thomas Mathivet<br>Holger Gerhardt |
| Stichting Tegen Kanker | Project Grant 2012-181 | Fabio Stanchi<br>Thomas Mathivet<br>Holger Gerhardt |
| European Molecular Biology Organization | Post-doc Fellowship | Thomas Mathivet |
| Fondation Leducq | ARTEMIS | Martin L Jones<br>Katie Bentley<br>Holger Gerhardt |
| Beth Israel Deaconess Medical Center | | Katie Bentley |
| Lister Institute of Preventive Medicine | Prize fellow | Holger Gerhardt |

The funders had no role in study design, data collection and interpretation, or the decision to submit the work for publication.

## Author contributions

BU, RAB, Designed experiments, Performed experiments, Wrote the manuscript, Analysis and interpretation of data; IG, FS, TM, Performed experiments, Conception and design, Analysis and interpretation of data; MLJ, Acquisition of data, Analysis and interpretation of data, Contributed unpublished essential data or reagents; AR, Performed experiments, Analysis and interpretation of data; KB, Conceived the idea of endothelial synchronization, and developed the computational model, Designed experiments, Wrote the manuscript, Acquisition of data, Analysis and interpretation of data; HG, Designed experiments, Wrote the manuscript, Analysis and interpretation of data

## Author ORCIDs

Martin L Jones, http://orcid.org/0000-0003-0994-5652
Holger Gerhardt, http://orcid.org/0000-0002-3030-0384

## Ethics

Animal experimentation: Mice were maintained at London Research Institute under standard husbandry conditions. All protocols were approved by the UK Home Office (P80/2391). Glioblastoma studies were performed at the Vesalius Research Center, VIB, KU Leuven where housing and all experimental animal procedures were performed in accordance with Belgian law on animal care and were approved by the Institutional Animal Care and Research Advisory Committee of the K U Leuven (P105/2012).

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
