## [Decision Letter]

Thank you for submitting your work entitled "Collective endothelial Dll4/Notch dynamics switch between vessel branching and expansion" for consideration by *eLife*. Your article has been reviewed by three peer reviewers, and the evaluation has been overseen by Tanya Whitfield as Reviewing Editor and Fiona Watt as the Senior Editor.

The reviewers have discussed the reviews with one another and the Reviewing Editor has drafted this decision to help you prepare a revised submission.

Summary:

This is an outstanding paper from a world-leading group presenting a new paradigm for consideration of how the vessels choose to branch or expand their diameter, which seem to be reciprocal processes. Through a series of new models, elegant microscopy and video microscopy, plus predictive computational modelling, this paper shows that Δ-like 4 fluctuates in individual endothelial cells in sprouting vessels. The authors propose a model by which the synchronisation (or lack thereof) of Dll4 expression between ECs governs their collective cell behaviour during vessel branching and expansion. The study proposes that under normal conditions, lateral inhibitory effects of Notch promote a salt and pepper expression of Dll4 that co-ordinates endothelial migratory dynamics to regulate angiogenesis, while under high VEGF conditions, Dll4 expression becomes synchronised in ECs, which disrupts normal sprouting leading to reduced branching and elongation. The switch between branching and expansion is a potentially critical phase in the development of blood vessels, highly relevant to understanding pathological angiogenesis and indeed, in the future, therapies.

Essential revisions:

1) Reviewer 3 has commented that the length of the manuscript and the sheer volume of data presented make it unwieldy for the reader. The format of *eLife* allows flexibility in publishing longer manuscripts, but it would be helpful if the authors could ensure that the manuscript is written as concisely as possible.

2) The zebrafish model, although elegant, would need further analysis for demonstration that branching behaviour is due to altered Dll4 signalling. This is perhaps an area that could be cut from the current MS and used to form the basis of a separate study, with further experiments as suggested by Reviewers 1 and 2.

3) If Dll4-blocking antibodies and the LLC lung carcinoma model are available, the experiments suggested by Reviewer 3 should be attempted.

The full reviews are appended for your information.

Reviewer #1:

It is clear the GBM staining data is of interest, showing the extensive empty collagen 4 sleeves, supporting the evidence for sprouting and retraction. Such sleeves have been seen before, however, with other extra-cellular proteins secreted by endothelial cells the work of Donald McDonald has shown empty sleeves of collagen after regression of vessels and then regrowth down these sleeves.

Also, whether the vessels that show little or no signal have full flow, needs to be considered.

It would have been useful to have an injection of antibody staining vessels to show perfusion to match the expression patterns of Δ-like 4, and to consider the possibilities that poor flow or vessel shutdown and shunting has caused some of these changes as opposed to dynamic fluctuations of Δ-like 4.

The zebrafish model is elegant and extremely well filmed. However, the retraction and sprouting, although clearly demonstrated, is not proven to be related to the Δ-like 4 signalling in this model. Is it blocked by notch inhibition and is it possible to do ISH in these experiments to look at the pattern of Δ-like 4 RNA expression?

The retinal experiments are highly convincing but it would be useful to have more validation in the brain tumour and zebrafish as these are perhaps the two most critical relevant to human therapy currently.

It would be useful to consider what the mechanism actually is that leads from the switch from branching to expansion. Is it possible, although obviously well beyond the scope of this article, it might be useful to discuss how this might be occuring. What might be downstream of Δ-like 4 involved in this pathway?

Reviewer #2:

The authors present strong evidence for local Notch-dependent synchronisation of Dll4 dynamics in vitro and in vivo by comparison of stabilised and destabilised fluorescent Dll4 reporters. The authors nicely demonstrate that the mitogenic functions of VEGF are not responsible for vessel expansion, which is surprising and lends further support to their hypothesis. The authors also highlight the existence of synchronous Dll4 dynamics during particular pathological angiogenic situations, (retinopathy and tumour angiogenesis) suggesting that Dll4 synchronisation may be central to EC pathology in these. However, the link between Dll4 dynamics and promotion of vessel expansion/disruption of branching is correlative and the authors do not formally demonstrate that Dll4 synchronisation causes the phenotypic changes. These experiments however, would be exceptionally technically demanding, if at all possible, and it is not the intention of this reviewer to request them. However, the authors may want to tone down statements implying evidence of a causal link between the phenomenon and the phenotype.

That said, these studies have important implications for the field by stimulating a reinterpretation of the idea that VEGF levels, rather than shifting the balance between tip cell migration and stalk cell proliferation to reduce branching and drive vessel expansion, may in fact act to control synchronisation of adjacent cells via Dll4.

The difficulties in defining individual cell boundaries during timelapses of embryoid body cultures (Figure 4) could have been circumvented by performing complementary experiments in zebrafish embryos e.g. by employing CRISPR-mediated knock in or established dll4 enhancers to drive destabilised reporters (Sacilotto et al., 2013). This may have also helped formalise the concept of Dll4 dynamics as the driver in this process because it would facilitate mosaic analysis of adjacent ECs with different competencies for receiving VEGF and inducing Dll4. That said, these experiments are technically challenging and may only serve to 're-invent the wheel'. Figure 4 is also particularly intractable and given its importance in the manuscript would benefit from some simplification to make the observations clearer.

Reviewer #3:

The manuscript is a technological tour de force that is very heavy for the reader. Interesting and important transgenic reporters are introduced and there a respectable effort to create a refined computational model to describe the endothelial responses to VEGF-Dll4/Notch synchronization. The data fits to the mechanistic explanation of the sprouting/branching and loss of polarization/expansion that started to evolve after the 2003 JCB paper of Dr. Gerhardt. The manuscript contains a wealth of data.

I have the following brief comments:

The manuscript should be shortened by at least one third.

The authors do not show that Dll4 is directly responsible for the pathological angiogenesis in the retina. This could be done by injection of Dll4 blocking antibodies. Would this inhibit the clustering of cells with high and low Dll4 expression?

It is somewhat disappointing the reporter mice did not work for the tumor transplants, and instead the tumor experiment employed a mouse-zebrafish hybrid model. It is not clear to what extent the results can be generalized. The architecture of tumor vessels is known to differ greatly between for example B16 melanomas and LLC lung carcinomas in mice. I would like to suggest that the authors include analysis of the LLC model, which is a simple experiment. Some discussion should also be included on Dll4-Notch signaling in vessel co-option, which is a common invasive growth pattern in the brain.

---

## [Author Response]

Essential revisions:

1) Reviewer 3 has commented that the length of the manuscript and the sheer volume of data presented make it unwieldy for the reader. The format of eLife allows flexibility in publishing longer manuscripts, but it would be helpful if the authors could ensure that the manuscript is written as concisely as possible.

We have shortened the manuscript according to the editorial and referee recommendations.

*2) The zebrafish model, although elegant, would need further analysis for demonstration that branching behaviour is due to altered dll4 signalling. This is perhaps an area that could be cut from the current MS and used to form the basis of a separate study, with further experiments as suggested by Reviewers 1 and 2.*

We have taken out the zebrafish study to facilitate shortening, and agree that this may be better published in a separate study. See detailed comments to this effect below.

*3) If Dll4-blocking antibodies and the LLC lung carcinoma model are available, the experiments suggested by Reviewer 3 should be attempted.*

We appreciate these suggestions. However, although these experiments would in principle be possible, we are not convinced they provide important additional information. We clearly show in the EB model that the clustering effect and synchronization is Notch dependent. The requested Dll4 antibody experiment in the retinopathy model has been performed (although with g-secretase inhibition) in previous work, cited below. It is well established that inhibiting Notch signaling in this context promotes branching. However, the underlying mechanism was not understood. At our new location in Berlin, after moving from London, we still don’t have the licence to perform the retinopathy and injection experiment.

Please also see our detailed response to the tumour model below.

Reviewer #1:

It is clear the GBM staining data is of interest, showing the extensive empty collagen 4 sleeves, supporting the evidence for sprouting and retraction. Such sleeves have been seen before, however, with other extra-cellular proteins secreted by endothelial cells the work of Donald McDonald has shown empty sleeves of collagen after regression of vessels and then regrowth down these sleeves.

We agree. It is however not clear to us whether this statement was intended as a suggestion for any changes to the manuscript.

*Also, whether the vessels that show little or no signal have full flow, needs to be considered.*

It would have been useful to have an injection of antibody staining vessels to show perfusion to match the expression patterns of Δ-like 4, and to consider the possibilities that poor flow or vessel shutdown and shunting has caused some of these changes as opposed to dynamic fluctuations of Δ-like 4.

We agree that changing flow conditions can impact on vessel morphogenesis, stability, diameter etc. This is well established, even if the underlying mechanisms are poorly understood. In the context of our present work, it is clear that the mechanism of differential dynamics versus synchronized dynamics are principally flow independent, as they occur in the EB sprouts that do not carry flow.

Also the retinal model illustrates that the morphogenic changes upon VEGF injection occur in perfused vascular networks. Similarly, it is well established that the tufts in OIR are perfused. Therefore it is clear that the observed mechanisms can occur in both unperfused systems including endothelial monolayers and perfused systems such as in vivo vascular networks. The tumour condition in particular is more tricky as our new dynamic imaging shows that flow patterns are not only diverse across the tumour but also change dynamically within given vessels. Therefore a true correlative analysis of flow changes and dll4 patterns would require the attempted but unsuccessful dynamic monitoring of both flow and dll4 expression changes with genetic reporters in vivo.

Whilst we would anticipate that such data give additional insights, and we assume that flow will have some impact on the dynamics, the present work identifies a basic mechanism that in its essence can function also independently of flow. This is essential for the understanding and any future discussion on flow influences should await new models and solid data.

The zebrafish model is elegant and extremely well filmed. However, the retraction and sprouting, although clearly demonstrated, is not proven to be related to the Δ-like 4 signalling in this model. Is it blocked by notch inhibition and is it possible to do ISH in these experiments to look at the pattern of Δ-like 4 RNA expression?

We agree that we do not provide direct evidence for the dynamic behavior in fish to be caused or related to dll4 signalling. In light of the overall comments and recommendations by the referees and editors, we have removed the zebrafish data and will follow up on this in a full new study in the future to further investigate the details. We are currently working on a new dynamic dll4 reporter in fish that should enable more detailed insights in this model. The omission of the current fish data allows shortening of the manuscript.

The retinal experiments are highly convincing but it would be useful to have more validation in the brain tumour and zebrafish as these are perhaps the two most critical relevant to human therapy currently.

We agree that more validation in the tumour and fish would be desirable, but this requires a set of new tools not currently available and satisfactory. Given the volume of data already included in the current work, we follow the Editors advice to shorten rather than add more at this point.

It would be useful to consider what the mechanism actually is that leads from the switch from branching to expansion. Is it possible, although obviously well beyond the scope of this article, it might be useful to discuss how this might be occuring. What might be downstream of Δ-like 4 involved in this pathway?

At this point we can only speculate. Future work will address this aspect in detail. However, our work on the role of VE-cadherin differential adhesion in angiogenesis (Bentley Nature Cell Biology 2014) identified differential VE-cadherin dynamics to operate downstream of differential Notch activity. The computational analysis of such a mechanism indicated that any loss of heterogeneous activity in Notch signaling would disrupt vessel elongation by synchronizing the cells and thus rendering them all equally adhesive. The idea from these studies would be that differential adhesion drives intercalation and thus vessel elongation, whereas synchronization disrupts intercalation, thus causing cells to cluster and thereby enlarge vessel diameter. We have now added a sentence in the Discussion of the revised version.

*Reviewer #2: The authors present strong evidence for local Notch-dependent synchronisation of Dll4 dynamics* in vitro *and* in vivo *by comparison of stabilised and destabilised fluorescent Dll4 reporters. The authors nicely demonstrate that the mitogenic functions of VEGF are not responsible for vessel expansion, which is surprising and lends further support to their hypothesis. The authors also highlight the existence of synchronous Dll4 dynamics during particular pathological angiogenic situations, (retinopathy and tumour angiogenesis) suggesting that Dll4 synchronisation may be central to EC pathology in these. However, the link between Dll4 dynamics and promotion of vessel expansion/ disruption of branching is correlative and the authors do not formally demonstrate that Dll4 synchronisation causes the phenotypic changes. These experiments however, would be exceptionally technically demanding, if at all possible, and it is not the intention of this reviewer to request them. However, the authors may want to tone down statements implying evidence of a causal link between the phenomenon and the phenotype.*

We greatly appreciate the positive assessment and enthusiastic support for our work. We also agree with the stated limitations, in particular that we do not directly show a causal link between synchronization of Dll4 and vessel expansion. However, we do demonstrate, at least in the EB system, that Notch signaling is required for the phenomenon of synchronization and vessel expansion. Together with the mechanistic studies on the differential adhesion and endothelial cell intercalation published previously (Bentley et al. NCB 2014), we feel there is sufficient evidence to suggest a causal link between synchronization and the expansion phenotype. We have taken care to avoid overstatements and hope this will be appropriately interpreted by the reader.

*That said, these studies have important implications for the field by stimulating a reinterpretation of the idea that VEGF levels, rather than shifting the balance between tip cell migration and stalk cell proliferation to reduce branching and drive vessel expansion, may in fact act to control synchronisation of adjacent cells via Dll4.*

*The difficulties in defining individual cell boundaries during timelapses of embryoid body cultures (Figure 4) could have been circumvented by performing complementary experiments in zebrafish embryos e.g. by employing CRISPR-mediated knock in or established dll4 enhancers to drive destabilised reporters (Sacilotto* et al.*, 2013). This may have also helped formalise the concept of Dll4 dynamics as the driver in this process because it would facilitate mosaic analysis of adjacent ECs with different competencies for receiving VEGF and inducing Dll4. That said, these experiments are technically challenging and may only serve to 're-invent the wheel'.*

We appreciate this comment and have been pursuing this goal also from the start. In fact, over long periods we discussed and collaborated with Sarah De Val to see whether the enhancers would be helpful. However, to date the dynamic response of these enhancers is not fully clear and we preferred to use the full BAC approach to be sure that we have the essential regulatory elements included. Our efforts to duplicate this approach in fish however were not successful. We are now working intensely on the CRISPR-mediated knock in but it will be some time before this is at a stage for analysis. We were hoping we may get somewhere during this revision period, but the knock-in is proving to be difficult (knock-out works well). Given the large amount of data already in this manuscript, and the editorial recommendation to take out our fish data to shorten, we suggest that the full dynamic analysis in vivo in fish is kept for a follow up study. As noted by the referee, it is not clear whether this will provide deeper insight or simply a confirmation.

Figure 4 is also particularly intractable and given its importance in the manuscript would benefit from some simplification to make the observations clearer.

We share the desire to keep this important figure simple. However, we also feel it is important to show original data rather than a schematic summary of the observations. We therefore chose to show representative examples of the three experimental situations, control, high Dll4 and high VEGF. The simplifications we came up with for ease of visualization are to show first all spheres for tracking, then just two at the front, and to show the tracking intensity profiles together, and separately on the right. We apologize for the small fonts and thin lines that have made it too difficult to read. We have now enlarged the fonts and lines to improve this figure.

Reviewer #3:

The manuscript is a technological tour de force that is very heavy for the reader. Interesting and important transgenic reporters are introduced and there a respectable effort to create a refined computational model to describe the endothelial responses to VEGF-Dll4/Notch synchronization. The data fits to the mechanistic explanation of the sprouting/branching and loss of polarization/expansion that started to evolve after the 2003 JCB paper of Dr. Gerhardt. The manuscript contains a wealth of data.

I have the following brief comments:

The manuscript should be shortened by at least one third.

We thank the referee for the enthusiastic and positive comments, and we appreciate the manuscript is rather dense and lengthy. We have worked very hard to make this complex topic and the diverse methodology used here understandable for a broad audience, and have continued this effort in the revision. Following the editorial advice and the comment by this referee, we shortened the text, although not quite by a third. All attempts to achieve this led to omission of critical information and loss of congruence. We feel the revised version is sufficiently concise but comprehensive.

The authors do not show that Dll4 is directly responsible for the pathological angiogenesis in the retina. This could be done by injection of Dll4 blocking antibodies. Would this inhibit the clustering of cells with high and low Dll4 expression?

We agree, we would have liked to do this, but Dll4 blocking antibodies were not available to use at the time. Also during our revision period, we did not have access to Dll4 blocking antibodies, but more importantly we did not have the license to perform this experiment at our new location in Berlin. This licence is pending. In any case, it is clear that blocking Notch signaling in the oxygen induced retinopathy model leads to increased branching and less tuft formation. This was already published in supplementary material in our original Nature paper on the role of Dll4 in tip cell biology in 2007 (Hellström et al. Nature 2007). However, the mechanisms we identify now, and the dynamic understanding was missing at that time.

*It is somewhat disappointing the reporter mice did not work for the tumor transplants, and instead the tumor experiment employed a mouse-zebrafish hybrid model. It is not clear to what extent the results can be generalized. The architecture of tumor vessels is known to differ greatly between for example B16 melanomas and LLC lung carcinomas in mice. I would like to suggest that the authors include analysis of the LLC model, which is a simple experiment. Some discussion should also be included on Dll4-Notch signaling in vessel co-option, which is a common invasive growth pattern in the brain.*

We agree with the referee. For more than a year we have now continuously invested heavily into getting the reporter mice working for live imaging in the tumour model. We invested 1.2M Euro into a 2P-LSM with the most sensitive detectors available, and optimized together with physicists and the company every aspect to get highest possible sensitivity. We can study angiogenesis in the tumour with great precision (manuscript without any Dll4/Notch data is currently in revision in science translational medicine, can be provided if requested) but sadly have not succeeded in detecting the destabilized reporter signal. At this time we cannot be sure about the reason. One confounding factor is the wavelength profile of Venus, which falls right in the middle of the autofluorescence spectrum. We get quite some signal in the expected Spectrum, but this is unfortunately not derived from the reporter. The attempts to separate any signal from autofluorescence that were successful in the ES cells, failed in the in vivo setting in the brain. Secondly, it is possible, that the overall levels in the brain are not as high as in the retina where we can clearly detect the signal when studying fresh material. This has been frustratingly tricky, but cannot be helped. Therefore the best direct data to date remain from the ES cell model. We hope that future work with new reporters including in the fish model will allow live imaging in living species including tumours. Importantly, we do not believe the lack of this in vivo data at this time invalidates our novel conclusions.

As for the tumour models, we disagree that comparing different tumour models, would provide new insight. Even if the architecture is different in different tumour models, how would this help? The important question is to see whether the different tumour vessel architecture that forms in different regions within a tumour correlates with evidence for altered Dll4 spatial patterns that would indicate possible breakdown of salt-and-pepper patterns. The combined Dll4 protein staining and ISH on tumour material is very challenging and lengthy procedure, which requires optimization for every new tissue type. We strongly feel that this investment is not justified given questionable insight prospects. In addition, confirming data in models that are intrinsically more variable and not orthotopic seems difficult to justify. Given that we are asked to reduce and shorten, we hope this referee will agree that we do not follow this avenue.

Concerning cooption, we agree this is a common mode in invasive tumours including glioblastoma. However, the model we are using is based on implantation of pre-aggregate tumour cell spheres. This model shows less invasive behavior at the borders and grows much more as a solid mass that attracts blood vessels. Having said that, there is some invasion at the borders, and we assume also some cooption. We would be happy to discuss this in relation to dll4/Notch if we had a good way to distinguish these modes in fixed tissues. At this time, any such discussion would be purely speculative and we don’t have any indication of specific differences or commonalities between coopted or sprouting-derived vessels. We suggest this is left out for future studies.